

**Wintertime Aerosol Measurements during the Chilean Coastal Orographic**
**Precipitation Experiment**
Sara Lynn Fults [1]
Adam K. Massmann [2]
Aldo Montecinos [3]
Elisabeth Andrews [4,5]
David E. Kingsmill [5]
Justin R. Minder [2]
Rene´ D. Garreaud [6]
Jefferson R. Snider [1,7]
[1] University of Wyoming, Laramie, WY
[2] University at Albany, Albany, NY
[3] Universidad de Concepción, Concepción, Chile
[4] NOAA ESRL Global Monitoring Division, Boulder, CO
[5] University of Colorado, Boulder, CO
[6] Universidad de Chile, Santiago, Chile
[7] Corresponding Author



## Abstract

The Chilean Coastal Orographic Precipitation Experiment (CCOPE) was a three-month
field campaign (June, July and August 2015) that investigated wintertime coastal rain events.
Reported here are analyses of aerosol measurements made at a coastal site during CCOPE. The
aerosol monitoring site was located near Arauco, Chile. Aerosol number concentrations and
aerosol size distributions were acquired with a Condensation Particle Counter (CPC) and an
Ultra High Sensitivity Aerosol Spectrometer (UHSAS). Arauco CPC concentrations were
compared to those measured at the NOAA observatory Trinidad Head (THD) on the North
Pacific Coast of California. The winter averaged CPC concentration at Arauco is 2971 cm$^{-3}$ $\pm$
1802 cm$^{-3}$; at THD the average is 1059 cm$^{-3}$ $\pm$ 855 cm$^{-3}$. Despite the typically more pristine
Southern Pacific region, the Arauco average is larger than at THD ($p < 0.01$). Aerosol size
distribution measurements acquired during episodes of onshore flow were analyzed with Köhler
theory and used to parameterize cloud condensation nuclei activation spectra. In addition, sea
salt aerosol (SSA) concentration was parameterized as a function of sea surface wind speed. It is
anticipated these parameterizations will be applied in modeling of wintertime Chilean coastal
precipitation.



## 1 Introduction

Forecast error due to incomplete understanding of atmospheric aerosols is evident in the predictions of many atmospheric models. As an example, general circulation models (GCMs) are used to forecast the Earth system's response to emissions of both aerosols and greenhouse gases. In spite of several decades of GCM development, the effect of aerosols on future climate remains uncertain (Boucher et al. 2013), particularly when compared to the greater certainty in climate forcing from anthropogenic greenhouse gases (e.g., Hansen 2009, his Fig. 10).

Aerosols perturb the abundance of cloud droplets and rain drops within clouds warmer than 0 ºC (liquid-only clouds). Via this interaction, both upward reflection of solar radiation by cloud cover (Albrecht 1989), and upward reflection by individual cloud elements (Twomey 1974) increase with increased aerosol abundance. Commonly referred to as aerosol indirect effects on climate, these processes decrease the amount of solar energy absorbed by the Earth system, and thus oppose global warming due to greenhouse gases. Other aerosol indirect effects, for example those due to aerosols nucleating ice in mixed-phase clouds (McCoy et al., 2014), augment greenhouse gas warming.

Because of its lower population and lower intensity of anthropogenic aerosol emissions, the Southern Hemisphere has been explored as a region for conducting studies of aerosol indirect effects and for exploring contrasts with the Northern Hemisphere (Schwartz, 1988; Gras, 1990; Gras 1995; Yum and Hudson 2004). This study contributes to those previous wintertime investigations of Southern Hemispheric aerosols. We emphasize the following topics: 1) The parameterized relationship between sea salt aerosol (SSA) concentration and sea surface wind speed; 2) The concentration of aerosol particles that are both smaller and more numerous than the SSA, and their role as cloud condensation nuclei (CCN); 3) The parameterized relationship



describing CCN activation spectra (Rogers and Yau, 1989; chapter 6), and 4) the potential
application of the SSA and CCN parameterizations in numerical modelling of wintertime
Southern Hemispheric clouds and precipitation. Motivating our investigation are modeling
studies (Feingold et al. 1999), and analyses of field measurements (Gerber and Frick 2012),
indicating that the reduction of rainfall due to increased CCN can be negated by SSA particles.

Measurements made with a Condensation Particle Counter (CPC), an instrument that

reports the concentration of particles with diameter ($D$) larger than ~ 0.01 μm, have formed the
basis of many previous investigations of aerosol abundance (Gras 1990; Brechtel et al. 1998;
Dall'Osto et al. 2009; Andreae 2009). These studies also evaluated air parcel back trajectories
and demonstrated that marine source regions are characterized by distinctly smaller CPC
concentrations than continental regions. Measurements of aerosol size distributions (ASDs) can
also aid understanding of the contrast between marine and continental CPC concentrations
(Brechtel et al. 1998; Birmili et al. 2001; Raes et al. 1997). The latter studies investigated
accumulation mode particles, centered at ~ 0.1 μm, and particles sizing in a mode at a distinctly
smaller central diameter (~ 0.05 μm). This smaller mode is commonly referred to as the Aitken
mode. In marine settings, the coexistence of both modes has been attributed to in-cloud
conversion of gas-phase sulfur dioxide ($SO_2$) to aerosol-phase sulfate (Hoppel et al. 1994), to
coalescence scavenging occurring within clouds (Hudson et al. 2015), and to new particle
formation (Covert et al. 1992; Petters et al. 2006). The latter process occurs in environments with
sufficiently enhanced ratios of $SO_2$ relative to aerosol.

The present work is an analysis of CPC concentrations and ASDs measured at a coastal

site on the Central Chilean Pacific coast during the Southern Hemisphere winter (June, July, and
August). Aerosol measurements were made during the Chilean Coastal Orographic Precipitation



Experiment (CCOPE) of 2015. CCOPE investigated aerosol properties and coastal orographic
precipitation and meteorology (Massmann et al. 2017).
This paper is organized into the following sections: Section 2 has descriptions of the
aerosol and meteorological instruments used to make surface measurements during CCOPE, and
Sect. 3 describes our analysis methods. Section 4 includes four topics: 1) Analysis of CPC
measurements and comparison to Coastal North Pacific measurements, 2) development of a
relationship between size-integrated aerosol concentration and size-integrated aerosol volume,
and comparison to similar relationships derived for summertime stratocumulus regimes, 3)
development of a parameterization of CCN activation spectra, and 4) development of a
parameterization of SSA number concentration. In Sect. 5, we compare our findings to previous
work, and in Sect. 6 we conclude with an outlook for how our parameterizations could be applied
in modeling of wintertime Central Chilean Pacific coast clouds and precipitation.
**2 Measurements**
**2.1 Measurement Site**
During CCOPE, a CPC (model 3010; TSI 2000a) and an Ultra High Sensitivity Aerosol
Spectrometer (UHSAS) (DMT 2013) were operated at a residence (37.25° S, 73.34° W, 55 m
above mean sea level (MSL)) near Arauco, Chile (population 35,000). Arauco is a coastal town
on the Central Chilean Pacific coast. Our measurement site, hereafter the Arauco Site (Fig. 1),
was selected because of our aim to characterize aerosols advecting onto South America from the
Southeast Pacific. Related to this, our measurements were coordinated with investigations of
rainfall inside the domain portrayed in Fig. 1. This study region lies in the South Pacific winter
storm track and rainfall here can be strongly enhanced by the Nahuelbuta Mountains (Garreaud



et al. 2016; Massmann et al. 2017). During CCOPE, several rainfall events were studied using
profiling radars and a precipitation disdrometer deployed at Curanilahue (Fig. 1), and a network
of precipitation gauges. The Arauco Site is located on a forested hill; most of the population of
Arauco lives east of the Arauco Site at an elevation less than 20 m MSL.

Salient characteristics of the CPC and UHSAS are provided in Table 1. These

instruments were operated inside the residence at the Arauco Site. In addition, a 3-meter
meteorological tower was deployed adjacent to the residence. Thermodynamic state (i.e., $T$, $P$,
and humidity) and horizontal wind speed and direction were measured on the tower. CPC and
meteorological measurements (minus wind direction) were acquired from 29 May to 14 August
(Table 1), UHSAS measurements were acquired from 29 May to 28 June (Table 1), and wind
direction measurements were acquired from 19 June to 14 August.

**2.2 Instrumentation**

Here we discuss characteristics of the CPC and UHSAS, sampling of the ambient

CCOPE aerosol, data acquisition of CPC and UHSAS measurements during CCOPE, and use of
the recorded UHSAS histograms to calculate ASDs. Additional information about the UHSAS is
provided in Appendix A. In that appendix we discuss how we validated, in a laboratory, the
UHSAS's determination of test aerosol concentration and particle size. During those validation
studies we intentionally dried the test aerosols to a relative humidity ($RH$) ≤ 15%. Consequently,
the effect of aerosol-bound water on either the physical size or the refractive index of the test
particles was negligible. UHSAS sizing of partially dried haze droplets ($RH$ ≤ 60 %), sampled
from the ambient atmosphere during CCOPE, and an associated particle size overestimate, is
also discussed in Appendix A. In Appendix A, we estimate the particle size overestimate to be ~

20 %.



134 During CCOPE, the CPC and UHSAS sampled ambient aerosol through a section of

135 copper tube (length = 3 m, inner diameter = 0.003 m, volumetric flow rate = 34 $cm^3$ $s^{-1}$). The

136 inlet end of the tube (hereafter, the sample tube) was secured below an eave on the west side of

137 the residence at the Arauco Site. The Reynolds number ($Re$) of the flow within the sample tube

138 was 960 and thus well below the value ($Re$ = 2300) where laminar flow changes to turbulent

139 flow. Particle transmission efficiencies were evaluated using Eq. (7.29) in Hinds (1999). These

140 are 78% for $D$ = 0.01 μm particles and ≥ 99% for $D$ = 0.1 μm and $D$ = 1 μm particles.

141 The CPC counts particles larger than $D$ = 0.010 μm (Table 1) [1] by detecting scattering

142 produced when aerosol particles are drawn through light emitted by a solid state laser ($λ$ = 0.78

143 μm). Prior to detection, the particle diameter is increased by at least a factor of 10 via alcohol

144 condensation. The aerosol sample flowrate in the CPC was 16 $cm^3$ $s^{-1}$. The CPC can detect a

145 maximum concentration of 10,000 $cm^{-3}$. During CCOPE, CPC concentrations were recorded

146 once per second and once every 10 seconds (Table 1).

147 The UHSAS measures scattering produced when aerosol particles are drawn through

148 light emitted by a solid state laser ($λ$ = 1.05 μm). By reference to a calibration table (Cai et al.

149 2008; Cai et al. 2013), the UHSAS microprocessor converts scattered light intensity to particle

150 size and accumulates the derived sizes in a 99 channel histogram. Channel widths are

151 logarithmically uniform ($\Delta log_{10}D$ = 0.013) over the instrument's full range (0.055 < $D$ < 1.0

152 μm). During CCOPE, the aerosol sample flow in the UHSAS was controlled at 0.34 $cm^3$ $s^{-1}$.

153 Eq. (1) was used to calculate the ASD.

---

[1] The CPC minimum detectable diameters we report are in fact diameters that a CPC detects particles with efficiency = 50 %. The CPC detection efficiency is a steep function of particle diameter (TSI 1996).



$$\left(\frac{dN}{d\log_{10} D}\right)_i = \frac{\Delta n_i}{\dot{V} \cdot \Delta t \cdot \Delta \log_{10} D} \qquad (1)$$

Here $\Delta n_i$ is the "i th" component of the count histogram and $\dot{V}$ is the aerosol flowrate. During
CCOPE, the UHSAS aerosol flow rate and the particle count histogram were recorded once
every ten seconds (Table 1), and hence, the sample interval ($\Delta t$ in Eq. (1)) is 10 s.
**3 Analysis**
**3.1 Air Mass Classification and Air Parcel Trajectories**
Locations close to the Arauco Site are shown in Fig. 1. A significant pollution source in
the region is the Arauco paper mill which releases 600 ton/yr of $SO_2$ (Arauco Woodpulp 2010).
When winds had an easterly component, the paper mill may have affected air quality at the
Arauco Site. Other pollution sources are Curanilahue (population 32,000) and Concepción
(population 950,000; this includes several municipalities adjacent to Concepción). In addition,
many residences in the region, including the residence where we operated the CPC and UHSAS,
burn wood for residential heating.
In a subsequent section, we compare CPC concentrations from the Arauco Site to values
measured at NOAA's Trinidad Head (THD) observatory in Northern California (41.05º N, 124.2º
W, 107 m MSL). The THD dataset includes contamination from local sources (e.g., campfires lit
by day visitors at the Trinidad State Beach Picnic Ground). Additionally, Mckinleyville, CA
(population 15,000) and Arcata, CA (population 18,000) are the two coastal population centers
reasonably close to THD. Both are southeast of the THD, at distances between 15 and 25 km.
Northern California's large population centers (San Francisco Bay Area and Sacramento) are ~
300 km southeast of the THD. An important distinction between the sampling at THD and



Arauco is the above ground level (a.g.l.) height of the aerosol inlets. This is 10 and 2 m a.g.l. at
THD and Arauco, respectively.

Wind measurements made at the Arauco Site (Sect. 2.1) and the THD were used to

conditionally sample the CPC measurements. At Arauco, wind directions from 180° to 330°
were chosen as the clean sector. At THD, the clean sector was chosen from 210° to 360°. The
clean sectors at Arauco and THD are shown in Fig. 2. Three factors entered into our selection of
the clean sectors: 1) Inclusion of winds from either true south (Arauco Site) or true north (THD),
2) the same range of angles (150º) at both sites, and 3) exclusion of wind from the directions of
regional population centers.

Additionally, we used HYSPLIT back trajectories (NOAA 2016) to conditionally sample

Arauco Site aerosol measurements associated with onshore-moving air. The back trajectories
were initialized at 00, 06, 12, and 18 UTC. In addition to these static arrival times, trajectories
were calculated with the coordinates of the Arauco Site [2] and with wind fields from the Global
Data Assimilation System. The spatial resolution of the wind data is 0.5º. Position along a
trajectory was evaluated hourly. Trajectories that were over the ocean continuously for three
days before landfall, and had a direction within the clean sector one hour before arriving at
Arauco, were classified as "onshore" trajectories. There are 20 onshore trajectories that overlap
with the availability of CCOPE UHSAS measurements.

In subsequent sections, a set of 20 two-hour data segments, centered on the onshore

trajectory arrival times, are further analyzed. Appendix B describes the numerical filter we used
to derive the aerosol properties analyzed in Sect. 4.2, 4.3, 4.4, and 4.5. The filter attenuates
aerosol property variability occurring on time scales shorter than 100 s. We developed the filter

---

[2] Trajectory starting altitude was set at 60 m MSL (5 m above the Arauco site)



to remove narrow "spikes" in the concentration sequences (CPC and UHSAS) which seem to
have originated from local sources of aerosol pollution. The Supplementary Material has plots of
filtered aerosol properties corresponding to each of the 20 two-hour segments. Four of these
were impacted aerosol variability at scales larger than 100 s. In general, these features were not
attenuated by the numerical filter. In these instances we discarded (subjectively) portions of the
two-hour segment and retained a subset for the analyses conducted in Sect. 4.3, 4.4 and 4.5.

Trajectory altitude is important for determining the presence of SSA particles. Onshore

trajectories originating from relatively close to the sea surface, and thus classified as onshore
"sea surface" trajectories, were required to have pressures > 980 hPa over their three-day
advection to the Arauco Site. Eighteen of the 20 onshore trajectories were also sea surface
trajectories. An example of a sea surface trajectory is shown in Figs. 3a - b. The sea surface wind
speed ($U$), analyzed in Sect. 4.5, is the average of the six hourly trajectory speeds in the six-hour
window ending six hours before the trajectory arrived at the Arauco site. The averaging interval
is shown in Fig. 3b. Two onshore trajectories, classified as "aloft", had pressures substantially
smaller than 980 hPa over their three-day advection to the Arauco Site.
**3.2 Sea Salt Aerosol**

Correlated values of SSA concentration and sea surface wind speed are reported in many

publications. In a review of the topic, Lewis and Schwartz (2004; hereafter LS04) used a
particle's deliquesced wet size, evaluated at 80% relative humidity, to group SSA particles into
three size classes. In field studies conducted at a coastal site, Clarke et al. (2003) demonstrated
that particles sizing in the middle of LS04's small particle size class - those with a dry diameter
larger than 0.5 μm or a RH = 80% wet diameter larger than 1 μm – had a composition that was
dominated by sea salt (NaCl).



By restricting our focus to segments of the CCOPE data associated with sea surface

trajectories (Sect. 3.1), we will analyze UHSAS-derived concentrations of particles with $D > 0.5$
μm ($N_{>0.5}$) and will assume that this subset of the ASD corresponds to SSA particles. This lower-
limit size is a factor of two smaller than the $RH = 80\%$ diameter corresponding to the middle of
LS04's small SSA class. This is because we assumed that particle size decreased as the aerosol
stream warmed from its ambient temperature to the temperature of the UHSAS measurement.
Support for this assumption is provided in Appendix A.

### 227 3.3 Moments of the Aerosol Size Distribution

In our analysis, we calculated three moments of the UHSAS-measured ASDs. These are

the aerosol concentration ($N_{UHSAS}$), aerosol surface area ($S_{UHSAS}$), and aerosol volume ($V_{UHSAS}$).
We symbolize these moments as integrals (Eq. (2) – (4)).
$$N_{UHSAS} = \int (dN/dlog_{10}D) \cdot dlog_{10}D \qquad (2)$$
$$S_{UHSAS} = \pi \int D^2 (dN/dlog_{10}D) \cdot dlog_{10}D \qquad (3)$$
$$V_{UHSAS} = (\pi/6) \int D^3 (dN/dlog_{10}D) \cdot dlog_{10}D \qquad (4)$$

In these formulae the group $(dN/dlog_{10}D) \cdot dlog_{10}D$ represents the concentration of aerosol

particles with diameter between $log_{10}D$ and $log_{10}D + dlog_{10}D$. Hence, when plotted versus the
logarithm of particle diameter, the area under the $dN/dlog_{10}D$ curve is proportional to the size-
integrated concentration. This is demonstrated in Figs. 4a – b where the size-integrated
concentration ($N_{UHSAS}$) is ~ 300 cm$^{-3}$ in onshore-moving air (Fig. 4a), and the concentration is
approximately four times larger (~ 1100 cm$^{-3}$) in air thought to be contaminated by continental
sources (Fig. 4b). Also apparent is the right-tail of an Aitken mode, at ~ 0.06 μm in Fig. 4a
(onshore-moving air), the absence of an Aitken mode in Fig. 4b (continental air), at least at
diameters detectable by the UHSAS ($D > 0.055$ μm; Table 1), and the presence of an





accumulation mode at ~ 0.1 μm in both airmasses (Figs. 4a – b). Two aspects of these results, i.e.
the absence of an Aitken mode plus the dominance of an accumulation mode, in polluted coastal
air, is consistent with ASDs reported in Raes et al. (1997) and in Dall'Osto et al. (2009).

**4 Results**

**4.1 Comparison of CPC Concentrations at Arauco Site and THD**

In this section, Arauco site CPC concentrations are compared to concentrations measured

at NOAA's THD observatory. At THD, concentrations were measured using a TSI 3760
condensation particle counter. The minimum particle diameter detected by the TSI 3760 ($D =$
0.011 μm; TSI 1996) is slightly larger than that in the TSI 3010 ($D = 0.010$ μm; Table 1). We
ignored this distinction.

The THD dataset spans the years 2002 to 2014. Because CCOPE was a wintertime field

study, only December, January, and February THD data are used in the comparison. There are
24,346 data points (hourly averaged) from THD and 5,541 classify as clean sector. In
comparison, there are 745 data points from the Arauco Site (hourly averaged) and 194 classify as
clean sector. For both sites we required a clean sector wind speed > 1.5 m s$^{-1}$ in addition to the
clean sector directional criteria (Fig. 2). Because the numerical filter (Sect. 3.1) requires 1 Hz
CPC measurements, and since 1 Hz measurements are unavailable in the THD data archive, the
filter was not applied to either of the data sets analyzed in this section.

In the following paragraph we compare hourly averages of CPC measurements from the

Arauco Site and THD. Because the number of data points in these data sets is different, a
particular statistical comparison methodology was applied. The approach followed here
compares the Arauco and THD average concentrations by applying the Student's t-distribution



method (t-test) explained in Havlicek and Crain (1988; their Eq. (10.6) and (10.7)). The
statistical hypotheses are: A) Null hypothesis: averages are equal, and B) Alternate hypothesis:
the averages are different. We also applied the non-parametric Wilcoxon Rank-Sum Test
(rs_test; Interactive Data Language, Harris Geospatial Solutions, Inc.). Statistical inference that
we derive based on the Wilcoxon Rank-Sum Test (not shown) is consistent with what we
describe below using the t-test.
Two aspects of the Arauco/THD comparison are presented here; more detail is available
in Fults (2016). First, clean sector concentrations are compared. The mean $N_{CPC}$ at Arauco is
2759 cm$^{-3}$ (standard deviation $\sigma$ = 1827 cm$^{-3}$). The mean and $\sigma$ at THD are 858 ± 729 cm$^{-3}$. Fig. 5
shows the Arauco and THD $N_{CPC}$ probability distribution functions. Of note is the larger mode
concentration and broader distribution at Arauco. Based on our t-test comparison, the Arauco
average is larger than the THD average ($p < 0.01$). Second, Arauco and THD concentrations are
compared without regard to wind direction. The average at the Arauco Site is 2971 cm$^{-3}$ ± 1802
while at THD the average is 1059 cm$^{-3}$ ± 855 cm$^{-3}$. These averages are also statistically different
($p < 0.01$), and again, the Arauco average is larger than that at THD. Furthermore, based on the
averages presented in this section, and the information presented in Table 2, we classify THD as
a moderately-polluted marine site, and the Arauco Site as between moderately-polluted marine
and heavily-polluted marine.
**4.2 Continental Contamination**
In this section we probe why aerosol properties varied strongly during four of the 20
onshore trajectories. Among these, the example presented in Figs. 6a – c exhibits the largest
degree of CPC and UHSAS variability. During this two-hour data segment, centered on 00 UTC
June 9 (9 pm local time), winds were light at Arauco and Curanilahue (< 2 m s$^{-1}$) and the wind





direction was variable at Curanilahue (Arauco Site wind direction measurements are only
available after 19 June 2015; Sect. 2.1).

Over the ocean, 12 to 6 hours prior to 00 UTC June 9, the HYSPLIT wind speed was 8.3

m s$^{-1}$ and the HYSPLIT direction was westerly (Fig. 3b). In terms of UHSAS measurements
(Figs. 6a – c), an obvious feature is the variability in the sequences of $N_{UHSAS}$, $V_{UHSAS}$, and $S_{UHSAS}$.
The $S_{UHSAS}$ is largest during an enhancement at ~ 00:37 UTC. The question arises: Can winds
over the ocean and the resultant SSA production cause this variability, or must continental
aerosol sources be evoked to explain this? This was addressed by calculating aerosol surface
areas as a function of wind speeds that bracket the HYSPLIT-derived wind speed (8.3 m s$^{-1}$). The
basis for this calculation is the $S$-on-$U$ parameterization described in LS04 (their Fig. 22). The
calculation indicates that $S$ can range between 6 μm$^2$ cm$^{-3}$ ($U$ = 6.3 m s$^{-1}$) and 15 μm$^2$ cm$^{-3}$ ($U$ =
10.3 m s$^{-1}$). Since the upper-limit of the predicted variation is small compared to $S_{UHSAS}$ at ~
00:37 UTC (Fig. 6c), and at other times in Fig. 6c, and because the wind speed variation applied
in the calculation is an order of magnitude larger than the variation in the HYSPLIT-derived
wind speed (± 0.1 m s$^{-1}$), it is concluded that the aerosol enhancements seen in Figs. 6a – c are
not due to a wind speed increase over the ocean. Rather, we surmise that aerosols emitted by
continental Chilean sources were sampled during portions of the segment in Fig. 6. Vertical
dashed lines indicate the subset of the two-hour segment we picked (subjectively) as being
representative of onshore-moving air that was not affected, or only moderately affected, by
emissions from continental Chilean sources. However, we do not expect our conditional
sampling (based on HYSPLIT) and subjective picking (e.g., Fig. 6) to select aerosol properties
representative of pristine marine air. Rather, we view these strategies as way to isolate aerosol



properties associated with onshore-moving air that was less affected by continental sources
compared to the other portions of the CCOPE data set.

Portions of three other two-hour segments were also discriminated into a period of

onshore-moving air that was less affected by continental aerosols compared to an adjacent
portion (or portions) of the two-hour data segment. This is shown in the Supplementary Material.
Only measurements seen plotted between the vertical dashed lines in the Supplementary Material
are analyzed in Sect. 4.3, 4.4, and 4.5.
**4.3 Using N/V ratios to Parameterize Cloud Droplet Concentrations**

In this section we analyze the ASD moments defined by Eq. (2) and (4). These are

symbolized $N_{UHSAS}$ and $V_{UHSAS}$, respectively. The ratio of $N_{UHSAS}$ (aerosol concentration) and
$V_{UHSAS}$ (aerosol volume) – generically the *N/V* ratio - is of interest for several reasons. First,
models that evaluate exchange between a marine boundary layer (MBL) and an overlying free
troposphere (FT) successfully predict the *N/V* ratio in the MBL (van Dingenen et al., 2000;
hereafter VD00). Second, a value of the ratio can be derived by fitting measurements of *N* and *V*
(Hegg and Kaufman 1998, hereafter HK98). Third, aerosol mass loading, and thus an aerosol
volume corresponding to an assumed particle density [3], are relatively easy to evaluate. A method
routinely used to evaluate aerosol mass involves pulling aerosol-laden air through a filter and
evaluating the accumulated mass gravimetrically. Fourth, the product of an *N/V* ratio and an
ambient aerosol volume has been proposed for estimating cloud droplet concentrations in marine
stratocumulus clouds (HK98 and VD00).

---

[3] In the case of ambient particles containing hygroscopic materials, density values range between 1.5 and 1.8 g cm$^{-3}$ (McMurry et al. 2002)



HK98 used a passive cavity aerosol spectrometer probe (PCASP) to evaluate $N$, $V$ and the
$N/V$ ratio. Since the UHSAS counts down to a smaller diameter (0.055 μm) than the PCASP
(0.12 μm), it is expected that the $N/V$ ratios we derive using the UHSAS will be larger than those
in HK98. The main reason for this is that decreasing the lower-limit diameter increases $N$ more
than $V$ (VD00).
As in HK98, linear least-squares regression analysis with an equation of the form $Y = a \cdot X$
was used to derive $N/V$ ratios. Values of $N_{UHSAS}$ and $V_{UHSAS}$ entered into the regressions were
derived with the lower-limit diameter set at 0.055 μm (Table 3) and 0.12 μm (Table 4). The latter
allows comparison to $N/V$ ratios in HK98. Tables 3 and 4 show the ratios and the fact that all of
the Pearson correlation coefficients ($r$) are positive. With the exception of trajectories arriving at
12 UTC June 5 and 06 UTC June 8 (Table 3), and at 00 UTC June 9 (Table 4), all of the $N/V$
correlations are statistically significant at $p < 0.01$.
As expected, the average $N/V$ ratio in the fifth column of Table 3 ($417 \pm 297$ μm$^{-3}$) is
larger than that in HK98 ($223 \pm 76$ μm$^{-3}$). These averages are different at $p = 0.01$. Table 4 has
results based on the larger lower-limit diameter (0.12 μm). In that comparison, the Arauco $N/V$
ratio ($159 \pm 100$ μm$^{-3}$) does not differ significantly from HK98's (i.e., $p > 0.01$).
Application of the $N/V$ ratio to aerosol-cloud-precipitation modelling requires knowledge
of the aerosol volume, or alternatively, knowledge of the aerosol mass loading and the aerosol
particle density. The aerosol volume is then multiplied by an average $N/V$ ratio (e.g., the average
at the bottom of the fifth column of Table 4), and their product is taken to be the modelled cloud
droplet concentration (HK98 and VD00). This is straight forward, at least from the perspective of
incorporating an aerosol-induced cloud feedback into a simulation, but it suffers from requiring
additional information about the aerosol (aerosol volume). Because the UHSAS was unavailable





for much of CCOPE (Table 1), aerosol volume is also unavailable. Another drawback is the
implicit assumption that only aerosol particles larger than the lower-limit diameter (e.g., 0.12 μm
in Table 4) form cloud droplets.

### 4.4 Using ASD and $N_{CPC}$ Measurements to Parameterize CCN Activation Spectra

Andreae (2009) analyzed a set of aerosol concentration measurements obtained from

collocated CPC and CCN instruments. Andreae's CPC measurements represent the concentration
of particles *no smaller than* a particular diameter (~ 0.01 μm; Table 1), and his CCN
measurements represent the concentration of particles that activate cloud droplets at a water
vapor supersaturation (*SS*) *no larger than* a particular value (Rogers and Yau, 1989; chapter 6).
The latter is *SS* = 0.4 % in Andreae (2009).

Similar to the relationship between CCN concentrations at *SS* = 0.4 % and CPC

concentrations (Andrea, 2009; his Fig. 2), we now describe how CPC and UHSAS
concentrations from CCOPE can be used to develop functions that describe CCN activation
spectra. In our development, the independent variable is a CPC-measured aerosol concentration.
While only estimates, the activation spectra we obtain represent an important step toward
evaluating how CCN affected cloud and precipitation during CCOPE. We envision that this
assessment will be advanced when our activation spectra are used to initialize numerical models.

Our first step is to select a particle diameter, apply this as a lower-limit diameter in Eq.

(2), and divide the resultant size-integrated UHSAS concentration by the coincident CPC-
measured concentration. Two examples of this are presented in Figs. 7a – b where we define the
UHSAS-to-CPC concentration ratio as a *fractional aerosol concentration* (*FAC*). We symbolize
these as *FAC*(*D*=0.055 μm) (Fig. 7a) and as *FAC*(*D*=0.120 μm) (Fig. 7b). As is illustrated, a





*FAC* can be interpreted as the fraction of the aerosol population *no smaller than* the lower-limit
diameter at the left-edge of the gray shading.
In a second step we interpret a *FAC*'s lower-limit diameter as an upper-limit *SS*. We do
this by applying a value for the kappa hygroscopicity parameter, which we set at $\kappa = 0.5$, and by
applying the kappa–Köhler formula of Petters and Kreidenweis (2007, their Eq. (11)). This
transformation from lower-limit $D$ to upper-limit $SS$ converts the *FAC* in Fig. 7a to *FAC*($SS =$
0.41 %) and the FAC in Fig. 7b to *FAC*($SS = 0.13$ %). We also evaluated how a range of the
kappa parameter ($0.3 < \kappa < 0.7$) translates to a range of *SS*. Our upper-limit $\kappa$ comes from
airborne measurements made over the Southeast Pacific Ocean during summer (Snider et al.,
2017), and our lower-limit $\kappa$ is the value recommended by Andreae and Rosenfeld (2008) for
simulating aerosol indirect effects over continents.
The *FAC*s in Figs. 7a – b are two examples of the many available from CCOPE. We
derived averaged *FAC*s, corresponding to each of five $N_{UHSAS}(D)$ sets (corresponding to five
selected lower-limit diameters ($D = 0.055, 0.070, 0.095, 0.120,$ and $0.200$ μm)), by plotting
$N_{UHSAS}(D)$ versus $N_{CPC}$ and fitting the data with the equation $Y = a \cdot X$ where $Y = N_{UHSAS}(D)$, $X =$
$N_{CPC}$, and "$a$" is the averaged *FAC*.
Averaged *FAC*s are presented in the seventh columns of Tables 3 and 4 where we
symbolize these as *FAC*($D = 0.055$ μm) and *FAC*($D = 0.120$ μm), respectively. Correlation
coefficients presented in the eighth columns of these tables mostly exceed 0.5. By averaging over
each of the 20 onshore trajectories, and noting that four of these were limited to a time interval
shorter than the nominal two hours (Sect. 4.2 and Tables 3 and 4), we calculated the overall
averages presented at the bottom of the two tables. These overall averages are *FAC*($D = 0.055$
μm) $= 0.35 \pm 0.13$ (Table 3) and *FAC*($D = 0.120$ μm) $= 0.13 \pm 0.07$ (Table 4). This decrease of



the *FAC* results because a larger lower-limit *D* (Eq. (2)), implies a smaller $N_{UHSAS}(D)$, and thus a
smaller *FAC(D)*.

What we refer to as *ensemble-averaged FAC*s were calculated by combining $N_{CPC}$ and

$N_{UHSAS}(D)$ values from all of the onshore trajectories. The selected data pairs were fitted in the
manner discussed previously. In addition, upper and lower quartile values of the fitted slopes
were calculated by applying the technique of Wolfe and Snider (2012; their Fig. 4d). We
evaluated four ensemble-averaged *FAC*s corresponding to four selected diameters (*D* = 0.070,
0.095, 0.120, and 0.200 µm). The *FAC* at *D* = 0.055 µm was eliminated from this analysis
because Kupc et al. (2018) showed that UHSAS-measured concentrations, at *D* ≤ 0.070 µm, are
negatively biased. Results are presented as circles in Fig. 8 and vertical error bars represent the
quartile range. Values plotted on the abscissa correspond to the four diameters, each transformed
to an *SS* using the kappa–Köhler formula with κ = 0.5, and horizontal error bars extend from
most hygroscopic (κ = 0.7), at the left-most limit, to least hygroscopic (κ = 0.3), at the right-most
limit.

In Fig. 8 we used power laws of the form $FAC(SS) = C \cdot SS^{k}$ (i.e., the form commonly used

to parameterize CCN activation spectra (Twomey 1959)) to fit the points. The change in the
slope of the fit function, seen here at *SS* = 0.15%, seems consistent with analyses demonstrating
that in polluted marine cloud conditions, albeit during summertime, the exponent "*k*" in the
Twomey power fit function is ≥ 1 and ≤ 1 at *SS* < 0.1 % and *SS* > 0.1 %, respectively (Hudson
and Nobel 2014; data from the MASE project in their Fig. 1).

Our parameterized CCN activation spectrum (Fig. 8) is relevant to cloud-aerosol-

precipitation modeling for several reasons. First, some numerical models treat *SS* as a prognostic
variable and thus require initialization with a CCN activation spectrum (e.g., Khairoutdinov and





Kogan 2000). Similarly, some models initialize with a particle size-dependent ASD function and
use Köhler theory to derive a model-initializing CCN activation spectrum (e.g., Lebo et al.
2012). As described in these two references, these models initialize with a nonspecific CCN
activation spectrum. If those models were used to investigate wintertime clouds and precipitation
on the Central Chilean Coast, our parameterization could be applied as a CCOPE-specific
initialization. Second, since we have measurements of $N_{CPC}$ for the totality of CCOPE (Table 1),
and we have shown how an ensemble-averaged CCN activation spectrum can be developed with
$N_{CPC}$ as the input parameter – i.e. as $N(SS) = FAC(SS) \cdot N_{CPC}$ – our parameterization can be used
to estimate activation spectra for the complete CCOPE campaign. Third, model initiation with a
specific CCN activation spectrum, as opposed to initialization with a regime-dependent droplet
concentration (e.g., Thompson et al. 2004), is justified by sensitivities to cloud droplet activation
reported in several publications (Cooper et al. 1997; Hudson and Yum, 1997; Snider et al.,

2017).

An assumption implicit in our development is that particles were internally mixed within

each of the four particle size classes. This seems justified by our use of HYSPLIT to
conditionally sample (Sect. 3.1), and by the fact that the sampled airmasses were resident in the
marine boundary layer for hours to days while subject to a variety of processes (Brownian
coagulation and reactive uptake of $SO_2$, among others) that produce aerosols consistent with the
internal mixture assumption (Fierce et al. 2017). An aspect of the measurements also supports
the internal mixture assumption. Fig. 7b shows that number concentration corresponding to the
0.120 to 1 µm class is dominated by particles with diameters at the lower end of that class.
Hence, the contribution of freshly emitted SSA particles, generally thought to size at dry
diameters larger than 0.5 µm (Clarke et al. 2003; LS04), and with a κ = 1.2 (Berg et al. 1998), is



typically small. A different bias would result if particles with κ values smaller than the lower-
limit value (κ = 0.3) contributed significantly to an $N_{UHSAS}(D)$ class. Burning biomass is an
important source for such low-hygroscopicity particles (Carrico et al. 2005). Our conditional
sampling (Sect. 3.1), combined with our filtering of the CPC and UHSAS measurements (Sect.
3.1 and Appendix B), reduces this concern.

**4.5 Regression of $N_{>0.5}$ and Sea Surface Wind Speed**

As discussed in Sect. 3.2, $N_{>0.5}$ represents the concentration of particles larger than 0.5
µm. We now support our conjecture that particles grouped into the $N_{>0.5}$ subset are indeed SSA.
We do this by analyzing the correlation between $N_{>0.5}$ and sea surface wind speed ($U$). Sect. 3.1
explains how we used HYSPLIT to derive $U$.
Values of $N_{>0.5}$, corresponding to the 18 sea surface trajectories (Sect. 3.1), are plotted
against $U$ in Fig. 9. Linear least-squares regression analysis with a model equation of form
$\ln(N_{>0.5}) = \ln(N_o) + a_N \cdot U$ was used to derive the coefficients $N_o$ and $a_N$ (O'Dowd and Smith
1993; LS04). The fitted coefficients are $N_o = 0.15$ cm$^{-3}$ and $a_N = 0.38$ and the derived function
(black curve) is shown in Fig. 9. The dashed black curves represent the 95% confidence interval
(Romano 1977; his Eq. (4.2.3.f)). Also plotted (pink line) is the function derived by O'Dowd and
Smith (1993) during ship-based sampling for dried particles with diameter between 0.38 and
0.84 µm. Given that the O'Dowd and Smith (1993) function (their Fig. 7a) is associated with
statistical uncertainty comparable to what we estimate for our data set, we are only moderately
confident that the function we derived is a consequence of wind-generated SSA. Two caveats
require mentioning. First, a fraction of our data points (~ 25%) lie either above or below our
confidence interval (Fig. 9). Meteorology can contribute to this variability, as when sea surface
winds establish a SSA population, and the wind subsequently slacks, or speeds up, prior to



advection onto the continent. This is expected because the atmospheric residence time of $D \sim 0.5$
μm particles, in the absence of precipitation, is several days (LS04, p. 76). Also, our
unintentional sampling of particles generated over the continent is a concern. We have taken
steps to eliminate those sources of contamination (Sect. 3.1 and Appendix B), but our methods
are not foolproof.

**5 Discussion**

The measurements analyzed here are, to the best of our knowledge, the first to

characterize aerosol concentrations and aerosol size distributions on the Central Chilean Pacific
coast during winter. Since the measurement site was relatively close to a population center
(Arauco, Chile), and a $SO_2$ emitting paper mill, and because wood burning is an important source
of residential heat in this region, we suspect that our measurements are influenced by these land
sources. We mitigated against this by focusing on data collected during periods of onshore flow.
Additional steps were taken to minimize contamination from land-based aerosol sources. These
procedures are explained in Sect. 3.1, 4.2, Appendix B, and in the Supplementary Material.

A point of comparison is the summertime measurements reported in HK98. Their data

was collected during airborne sampling over the western Atlantic in air that had advected from
the United States. HK98's averaged aerosol surface area ($131 \pm 93$ μm$^2$ cm$^{-3}$; their Table 2) is
clearly larger than that for our 20 onshore trajectories ($42 \pm 27$ μm$^2$ cm$^{-3}$; results not shown).
However, a more relevant comparator would be low altitude measurements made off the Central
Chilean Pacific during winter. As far as we know, the desired data set is not available. Values of
aerosol surface area in the FT over the North and South Pacific are generally $< 10$ μm$^2$ cm$^{-3}$
(Clarke 1992), suggesting that even during onshore flow the Arauco Site measurements are



affected by anthropogenic sources. We have assumed these sources are Chilean, however, a
contribution from long range transport cannot be ruled out.

The larger wintertime-averaged CPC concentration at Arauco, compared to THD, is

evidence for stronger continental contamination at Arauco. Since $N_{CPC}$ is a parameter in our
parameterization of CCN activation spectra (Sect. 4.4), we conclude that cloud droplet
concentrations in low level marine clouds (stratocumulus) formed in the vicinity of Arauco are
larger than in similar clouds near THD.  If true, this conclusion would be opposite the general
situation in Southern Pacific boundary layer clouds where cloud droplet concentrations are
statistically less than in their Northern hemispheric counterparts (Bennartz 2007). Relevant to
this, Bennartz (2007) comments on a coast-normal droplet concentration gradient that is stronger
on the Central Chilean coast compared to the California/Oregon coast. We presume that the
gradient exists because of the larger concentration of aerosols over continents (Andreae and
Rosenfeld, 2008), and because of aerosol removal that occurs within and below marine
stratocumulus clouds. In addition, Bennartz (2007) demonstrates that the coast-normal droplet
concentration gradient is larger off the Central Chilean coast, compared to California/Oregon
coast, in part because oceanic concentrations, ~ 2000 km offshore, are generally smaller in the
south compared to the north Pacific. Whether the southern hemispheric gradient is also enhanced
by larger aerosol concentrations over coastal Central Chile, compared to coastal California and
Oregon, is an open question. Further analysis of the satellite retrievals analyzed by Bennartz
(2007), with segregation into wintertime and summertime categories, as well as measurements
conducted at an offshore island location, or acquired using aircraft or ships, are needed to resolve
this question.



## 6 Conclusions


Analyses presented here are based on Condensation Particle Counter (CPC)

measurements made during one winter season (June, July and August 2015) on the Central
Chilean Pacific coast (38 $^{\circ}$ S). Also analyzed are aerosol size distribution measurements made
with an Ultra High Sensitivity Aerosol Spectrometer (UHSAS). UHSAS measurements are
available from 29 May to 28 June (Table 1). Limitations of this study are proximity of the
measurement site to a population center (Arauco, Chile) and a $SO_2$ emitting paper mill, sampling
of particles emitted from residences close to where our instruments were operated, and the
incomplete drying of the sampled aerosol particles. This first attempt to make CPC and ASD
measurements on the Central Chilean Pacific coast during winter was exploratory and our results
should be considered preliminary.

We compared the Arauco Site CPC measurements to values acquired at the NOAA

observatory Trinidad Head (THD) on the North Pacific Coast of California. Averaged CPC
concentrations are larger at the Arauco Site and that difference is evident in Arauco/THD
comparisons based on air arriving from all wind directions and from clean sector directions. In
addition, we conditionally sampled the UHSAS measurements and derived parameterized
descriptions of sea salt aerosol (SSA) and cloud condensation nuclei (CCN) for periods of
onshore flow. In these parameterizations the input parameters are respectively sea surface wind
speed and CPC-measured aerosol concentration.

In the context of CCOPE, there are two precipitation regimes that impact the Central

Chilean Coast and the Nahuelbuta Mountains during winter (Massmann et al. 2017). The first of
these have radar-derived echo tops at ~ 2 km MSL and produce rain by direct conversion of
cloud droplets to rain drops. The second have higher echo tops, extending to temperatures colder



than 0 °C and produce rain that is, at least in part, initiated by ice phase processes. Investigation
of the rain produced in the shallow regimes is an active area research; it is thought that SSA and
the CCN play important roles (Feingold et al. 1999; Gerber and Frick 2012). The deep regimes
form precipitating hydrometeors (ice particles) at cloud temperatures < 0 °C. Again, aerosols
play a role, but there are many facets to this and first-order effects are not yet agreed on. Perhaps
foremost is the role played by aerosol acting as ice nuclei. Measurement of an ice nuclei
activation spectrum, development of an ice particle parameterization, and incorporation of the
parameterization into a numerical model are needed to explore this dimension of the problem.
Because they modulate cloud droplet size, the development of graupel, and influence latent
heating (e.g., Tao et al. 2012), the CCN and SSA likely also play a role in the deep regimes.
Thus, we anticipate that modeling of both precipitation regimes will benefit from the CCN and
SSA parameterizations presented here.



**Author Contribution**

Jeff Snider, Jason Minder, David Kingsmill wrote successful proposals that funded this research. Sara Fults, Adam Massman, Aldo Montecinos, and David Kingsmill performed the field measurements. Rene´ Garreaud and Aldo Montecinos provided logistical support during the field phase of the project. Elisabeth Andrews provided data from THD. Sara Fult wrote her MS dissertation and this was adapted to this manuscript by Jeff Snider. All authors contributed to the editing of this manuscript.

**Acknowledgments**

We thank Freddy Echeverría-Cabezas for his assistance during CCOPE, Matthew Burkhart for building the aerosol data acquisition system, Zhien Wang for providing a graduate assistantship, Nicholas Mahon for shipping logistics, and the Departamento de Geofísica at the Universidad de Concepción. This work was supported by the United States National Science Foundation Physical and Dynamic Meteorology Division under Awards AGS-1522277 and AGS-1522939.

**Data Availability**

CCOPE CPC and UHSAS data, and a data reader (Interactive Data Language, Harris Geospatial Solutions, Inc.), are at http://www-das.uwyo.edu/~jsnider/CCOPE/.



**Appendix A:**

Because the *RH* at the Arauco Site was often in excess of 80 % (Fig. A1c), particles

entering the sample tube (Sect. 2.2) were haze droplets (Rogers and Yau 1989). As these haze
droplets transit the sample tube they experience an increase in temperature, an *RH* decrease, and
thus a decreased *D*. The lowest *RH* experienced by a haze droplet is at the point of detection
where the aerosol temperature is presumed to be the internal "box temperature" measured by the
UHSAS. The *RH* at this point is

$$RH_U = \frac{RH_A \cdot e_s(T_A)}{e_s(T_U)}$$

(A1)

where $T_U$ is the internal UHSAS temperature, $e_s$ is saturation vapor pressure (temperature
dependent), and $RH_A$ and $T_A$ are the ambient *RH* and temperature, respectively. In nearly all of
the UHSAS sampling during CCOPE, the $RH_U$ was less than 60 % (Fig. A1d). This suggests that
the haze droplets detected by the UHSAS were partially dried. Partial drying of the haze droplets
is supported by thermodynamic calculations (Lewis and Schwartz 2004; their Fig. 8) showing
that a $D = 4$ μm NaCl haze droplet reaches its equilibrium size ($D = 2$ μm) in 0.1 s subsequent to
a step-change of *RH* from 98 % to 80 %. Because 0.1 s is small relative to the average residence
time of haze droplets within the sample tube (0.8 s), we ignored the possibility of a kinetic
limitation to drying and we assumed that the haze droplets relaxed to their equilibrium size at
$RH_U$ prior to the time they were detected. Since we do not know the chemical composition of the
haze droplets, their equilibrium size is not specifiable, but calculations corresponding to $RH_U =$
60% and a haze droplet composed of sodium sulfate indicate that the equilibrium size is 30%
larger than the corresponding dry particle size (Snider et al. 2017; their Fig. A2b). Three factors
interact to partially compensate for a size overestimate due to incomplete particle drying during



CCOPE: 1) Particle sizing performed by the UHSAS was calibrated using polystyrene latex
particles (refractive index $n = 1.57$ at $\lambda = 1.05$ μm (Marx and Mulholland 1983)); 2) Liquid water
($n = 1.32$ at $\lambda = 1.05$ μm (Irvine and Pollack, 1968)) makes a significant contribution to the mass
of a haze droplet at $RH = 60\%$ (here again we are assuming the above-mentioned sodium sulfate
composition for the completely dried particle); and 3) Assuming the same scattering intensity, an
$n = 1.6$ particle sizes 10% smaller than an $n = 1.4$ particle (Cai et al., 2008; their Fig. 1).
Accepting the 10% as an underestimate, and the above-mentioned 30% as an overestimate, we
conclude that particle sizes reported by the UHSAS were overestimated by 20%. We did not
correct for this sizing bias.

Laboratory testing of the UHSAS and CPC is documented in Figs. A2a – b, and in Figs.

A3a - b. We evaluated consistency among measurements made with the UHSAS, the CPC, and a
Scanning Mobility Particle Scanner (SMPS; TSI 2000b). In all of these tests, the $RH$ of the test
aerosols was < 15 %. An example ASD derived using the UHSAS (pink) and the SMPS (black)
is shown in Fig. A2a. In this test the three instruments (UHSAS, CPC and SMPS) were sampling
mobility-selected ammonium sulfate particles with $D = 0.075$ μm. The refractive index of this
material at $\lambda = 1.05$ μm is $n = 1.51$ (Toon et al., 1976). It is evident that the mode diameter
measured by the UHSAS is smaller than that reported by the SMPS ($D = 0.075$ μm). This
difference is qualitatively consistent with the smaller refractive index of the test material
(ammonium sulfate), compared to the larger refractive index of the polystyrene latex particles
used by the factory to calibrate the UHSAS (DMT, 2013). Fig. A2b shows a test with $D = 0.71$
μm polystyrene latex particles. As expected, the mode diameter in the UHSAS size distribution
is in agreement with the mode size in the SMPS size distribution.





An additional feature of our laboratory testing is the multi-modal structure in the SMPS
size distribution at $D < 0.5$ μm (Fig A2b). This structure results because the particle diameter
inferred by the SMPS depends on the physical diameter of the test particles, and on also depends
on the test particle's charge state. The multi-modal structure at $D < 0.5$ μm corresponds to
particles carrying 5, 4, 3, and 2 fundamental charges, but each with physical diameter equal 0.71
μm. As stated in the previous paragraph, the latter is the diameter of the polystyrene test
particles.
Figs. A3a - b summarize all of the lab testing we conducted in support of CCOPE. In Fig.
A3a, $N_{UHSAS}$ is plotted vs $N_{CPC}$ for tests with $D < 0.2$ μm and Fig. A3b has tests with $D > 0.2$ μm.
On average, concentrations differ by ± 6 % in Fig. A3a ($D < 0.2$ μm) and by ± 10 % in Fig. A3b
($D > 0.2$ μm).





**Appendix B:**

For each of the onshore trajectories (Sect. 3.1), a two-hour segment, centered on the trajectory arrival time was analyzed. An example is in Figs. B1a – e. Fig. B1a shows the sequence of CPC values sampled every second (i.e., 1-s samples referred to as *fast $N_{CPC}$*), and Fig. B1b shows CPC values sampled every 10 seconds (i.e., 10-s samples referred to as *slow $N_{CPC}$*). The following procedure was used to attenuate the narrow perturbations (e.g., within the time interval indicated by vertical dashed lines in Figs. B1a, B1b, and B1d) that were likely the result of local aerosol emissions.

First, the fast $N_{CPC}$ values were used to determine, for each 10 s of the sequence, a concentration relative standard deviation ($\sigma / <x>$). Second, if the relative standard deviation was greater than 0.02 both the slow $N_{CPC}$ measurement (sampled once every 10 second) and the ASD measurement (also sampled once every 10 second; Table 1) were discarded. Fig. B1c and Fig. B1e show the $N_{CPC}$ and $N_{UHSAS}$ sequences after application of the filter. These two filtered sequences ($N_{CPC}$(filtered) and $N_{UHSAS}$(filtered)), and the filtered values of aerosol surface area ($S_{UHSAS}$), aerosol volume ($V_{UHSAS}$), and $D > 0.5$ µm concentration ($N_{>0.5}$) are the focus of the bulk of our analysis.



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





Table 1. Aerosol Instruments

| Instrument and Reference | Aerosol Property Measured | Particle Diameter Range, μm | Aerosol Flow Rate, cm³ s⁻¹ | Data Acquisition Rate, Hz | Data Availability (2015) |
|---|---|---|---|---|---|
| CPC Model 3010 (TSI 2000a) | Aerosol Concentration | $D > 0.010$ | 17 | 1 and 0.1 | 29 May to 14 Aug |
| UHSAS (DMT 2013) | Aerosol Size Distribution | $0.055 < D < 1$ | 0.34 | 0.1 | 29 May to 28 June |







Table 2. Classification of Air Mass Type

| Citation and Location | Measurement Site Characteristics | Air Mass Classification | Averaged CPC Concentration, cm$^{-3}$ [a] |
|---|---|---|---|
| Gras (1990) Cape Grim, Tasmania 40.68 ºS; 144.7 ºE | Oceanic Wintertime | Remote Marine | 100 |
| Brechtel et al. (1998) Macquarie Island (Southwest Pacific) 54.50 ºS; 159.0 ºE | Oceanic Summertime | Remote Marine | 700 |
| Diesch et al. (2012) Portugal 37.11 ºN; 7.735 ºW | Coastal Continental Late Autumn | Moderately-polluted Marine Heavily-polluted Marine Continental | 1000 7000 10000 |
| This Study Arauco, Chile 37.25 ºS; 73.34 ºW | Coastal Continental Wintertime | Between moderately-polluted Marine and Heavily-polluted Marine | 3000 |
| This Study Trinidad Head, CA 41.05 ºN; 124.2 ºW | Coastal Continental Wintertime | Moderately-polluted Marine | 1000 |


[a] Values rounded to one significant digit



Table 3. Statistics for Onshore Trajectories ($D$ integration in Eq. (2) and (4) is from 0.055 to 1 µm)

| Arrival Hour, UTC | Type | Start DDHHMM [a], UTC | End DDHHMM [a], UTC | $N_{UHSAS}$ on $V_{UHSAS}$ Slope, µm$^{-3}$ | $r$ [b] | $FAC(D=0.055$ µm$)$ | $r$ [c] | Number of Samples |
|---|---|---|---|---|---|---|---|---|
| 06 | Sea Surface | 050500 | 050700 | 93. | 0.54 | 0.59 | 0.65 | 139 |
| 12 | Sea Surface | 051100 | 051134 | 64. | 0.10 | 0.19 | 0.59 | 63 |
| 18 | Sea Surface | 051700 | 051900 | 110. | 0.66 | 0.41 | 0.63 | 342 |
| 00 | Sea Surface | 052300 | 060100 | 298. | 0.81 | 0.51 | 0.96 | 316 |
| 06 | Sea Surface | 060500 | 060700 | 60. | 0.53 | 0.18 | 0.89 | 677 |
| 12 | Sea Surface | 061100 | 061300 | 91. | 0.60 | 0.16 | 0.65 | 647 |
| 18 | Sea Surface | 061700 | 061900 | 107. | 0.33 | 0.18 | 0.81 | 476 |
| 00 | Sea Surface | 062300 | 062325 | 234. | 0.81 | 0.36 | 0.97 | 133 |
| 06 | Sea Surface | 080500 | 080700 [d] | 163. | 0.06 | 0.29 | 0.52 | 542 |
| 12 | Sea Surface | 081100 | 081300 | 358. | 0.75 | 0.28 | 0.76 | 504 |
| 18 | Sea Surface | 081700 | 081900 | 450. | 0.88 | 0.42 | 0.90 | 416 |
| 00 | Sea Surface | 090020 | 090033 | 764. | 0.45 | 0.34 | 0.98 | 72 |
| 06 | Sea Surface | 090500 | 090700 | 703. | 0.68 | 0.23 | 0.96 | 554 |
| 12 | Sea Surface | 091100 | 091300 | 714. | 0.89 | 0.44 | 0.94 | 532 |
| 18 | Sea Surface | 091700 | 091900 | 675. | 0.78 | 0.39 | 0.53 | 592 |
| 00 | Sea Surface | 092300 | 100100 | 519. | 0.37 | 0.22 | 0.68 | 618 |
| 06 | Aloft | 100500 | 100700 | 857. | 0.96 | 0.39 | 0.82 | 617 |
| 18 | Sea Surface | 101700 | 101900 | 825. | 0.86 | 0.37 | 0.19 | 622 |
| 00 | Sea Surface | 110006 | 110031 | 834. | 0.96 | 0.50 | 0.99 | 61 |
| 00 | Aloft | 262300 | 270100 | 420. | 0.68 | 0.47 | 0.93 | 647 |
| | | | <x> | 417 | | 0.35 | | |
| | | | σ | 297 | | 0.13 | | |
| | | | σ / <x> | 0.71 | | 0.36 | | |


[a] DDHHMM indicates the start and end times (day in June 2015, hour, minute) of the data segment
[b] Pearson product moment for the $N_{UHSAS}(D=0.055$ µm$)$ on $V_{UHSAS}(D=0.055$ µm$)$ correlation
[c] Pearson product moment for the $N_{UHSAS}(D=0.055$ µm$)$ on $N_{CPC}$ correlation
[d] Data recording ended at DDHHMM = 080646, i.e., 14 min before the stated end time



Table 4. Statistics for Onshore Trajectories ($D$ integration in Eq. (2) and (4) is from 0.120 to 1 μm)

| Arrival Hour, UTC | Type | Start DDHHMM [a], UTC | End DDHHMM [a], UTC | $N_{UHSAS}$ on $V_{UHSAS}$ Slope, μm$^{-3}$ | $r$ [b] | $FAC(D=0.120$ μm$)$ | $r$ [c] | Number of Samples |
|---|---|---|---|---|---|---|---|---|
| 06 | Sea Surface | 050500 | 050700 | 60. | 0.74 | 0.37 | 0.47 | 139 |
| 12 | Sea Surface | 051100 | 051134 | 40. | 0.31 | 0.12 | 0.36 | 63 |
| 18 | Sea Surface | 051700 | 051900 | 64. | 0.76 | 0.23 | 0.49 | 342 |
| 00 | Sea Surface | 052300 | 060100 | 113. | 0.84 | 0.17 | 0.84 | 316 |
| 06 | Sea Surface | 060500 | 060700 | 34. | 0.67 | 0.10 | 0.78 | 677 |
| 12 | Sea Surface | 061100 | 061300 | 44. | 0.77 | 0.07 | 0.42 | 647 |
| 18 | Sea Surface | 061700 | 061900 | 42. | 0.61 | 0.06 | 0.24 | 476 |
| 00 | Sea Surface | 062300 | 062325 | 107. | 0.93 | 0.15 | 0.92 | 133 |
| 06 | Sea Surface | 080500 | 080700 [d] | 89. | 0.72 | 0.12 | 0.02 | 542 |
| 12 | Sea Surface | 081100 | 081300 | 139. | 0.79 | 0.09 | 0.53 | 504 |
| 18 | Sea Surface | 081700 | 081900 | 202. | 0.92 | 0.17 | 0.83 | 416 |
| 00 | Sea Surface | 090020 | 090033 | 184. | 0.12 | 0.06 | 0.78 | 72 |
| 06 | Sea Surface | 090500 | 090700 | 228. | 0.58 | 0.06 | 0.87 | 554 |
| 12 | Sea Surface | 091100 | 091300 | 262. | 0.92 | 0.14 | 0.73 | 532 |
| 18 | Sea Surface | 091700 | 091900 | 257. | 0.89 | 0.12 | 0.41 | 592 |
| 00 | Sea Surface | 092300 | 100100 | 204. | 0.83 | 0.06 | 0.32 | 618 |
| 06 | Aloft | 100500 | 100700 | 323. | 0.96 | 0.11 | 0.82 | 617 |
| 18 | Sea Surface | 101700 | 101900 | 279. | 0.91 | 0.10 | 0.08 | 622 |
| 00 | Sea Surface | 110006 | 110031 | 346. | 0.97 | 0.16 | 0.96 | 61 |
| 00 | Aloft | 262300 | 270100 | 171. | 0.65 | 0.18 | 0.88 | 647 |
| | | | <x> | 159 | | 0.13 | | |
| | | | σ | 100 | | 0.07 | | |
| | | | σ / <x> | 0.63 | | 0.55 | | |

[a] DDHHMM indicates the start and end times (day in June 2015, hour, minute) of the data segment
[b] Pearson product moment for the $N_{UHSAS}(D=0.120$ μm$)$ on $V_{UHSAS}(D=0.120$ μm$)$ correlation
[c] Pearson product moment for the $N_{UHSAS}(D=0.120$ μm$)$ on $N_{CPC}$ correlation
[d] Data recording ended at DDHHMM = 080646, i.e., 14 min before the stated end time



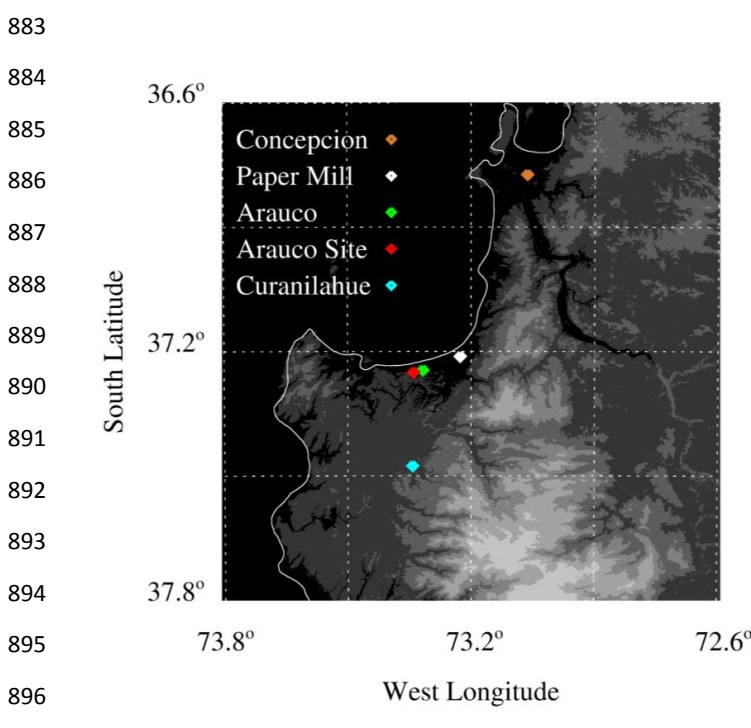

















Fig. 1 – Central Chilean Coastal region and the location of Arauco Site where aerosol
measurements were made during CCOPE. Altitude thresholds for the digital elevation map are at
0 m MSL, 50 m MSL, 250 m MSL, 500 m MSL, 750 m MSL, and 1000 m MSL.













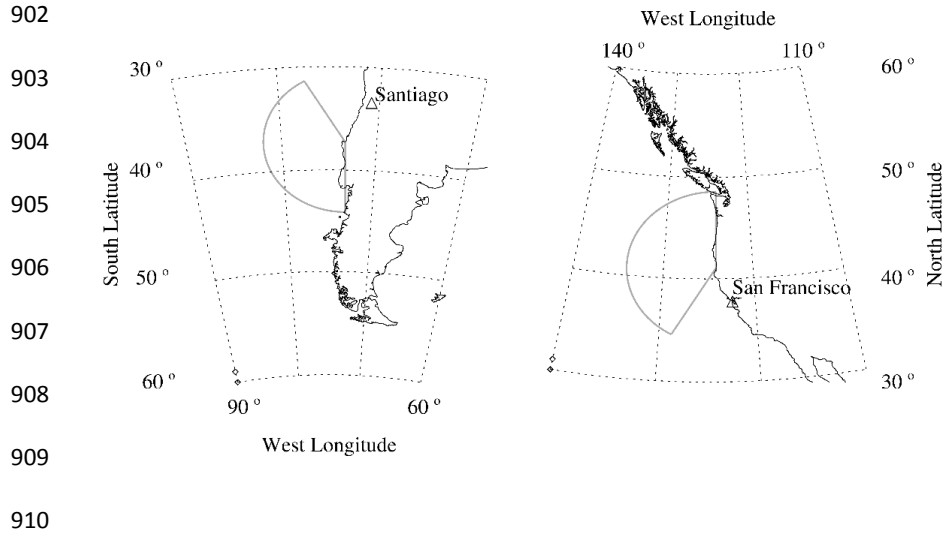



Fig. 2 - Clean sector chosen for Arauco (left, 180° to 330°) and the clean sector chosen
for THD (right, 210° to 360°).



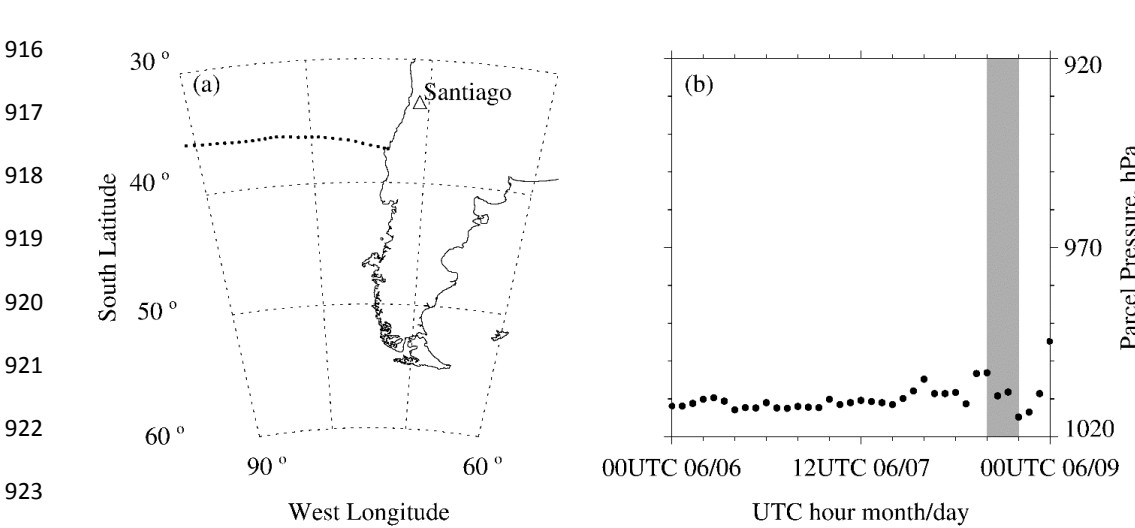

Fig. 3 - a) One of the 18 sea surface trajectories that arrived at the Arauco Site between 29 May to 28 June; this trajectory arrival occurred at 00 UTC June 9. Black dots are hourly output of the HYSPLIT model; however, for clarity, only every other 1-hr point is plotted. b) Hourly HYSPLIT pressure vs time; however, for clarity, only every other 1-hr point is plotted. The averaged sea surface wind speed ($U$) was evaluated over the 12 to 18 UTC interval show in gray.



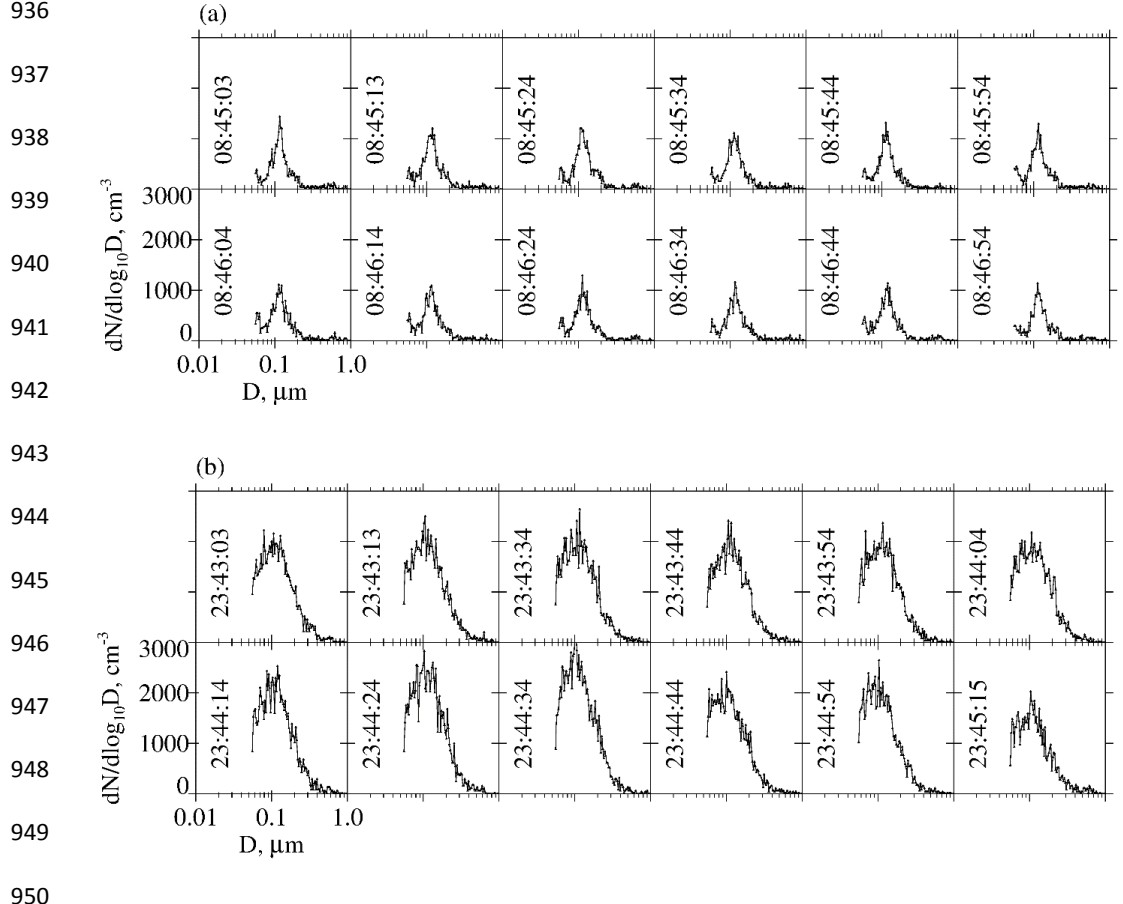

Fig. 4 - Consecutive ASDs recorded by the UHSAS at the Arauco Site. a) ASDs with a relatively small concentration (~ 300 cm$^{-3}$), a right tail of an Aitken mode (at ~ 0.06 μm), and an accumulation mode (at ~ 0.1 μm), in onshore-moving air on June 5, 2015. b) ASDs with a proportionately larger concentration (~ 1100 cm$^{-3}$), an accumulation mode (at ~ 0.1 μm), and no evidence of an Aitken mode, in air thought to be contaminated by continental sources (June 4, 2015). UTC time is written in each panel.




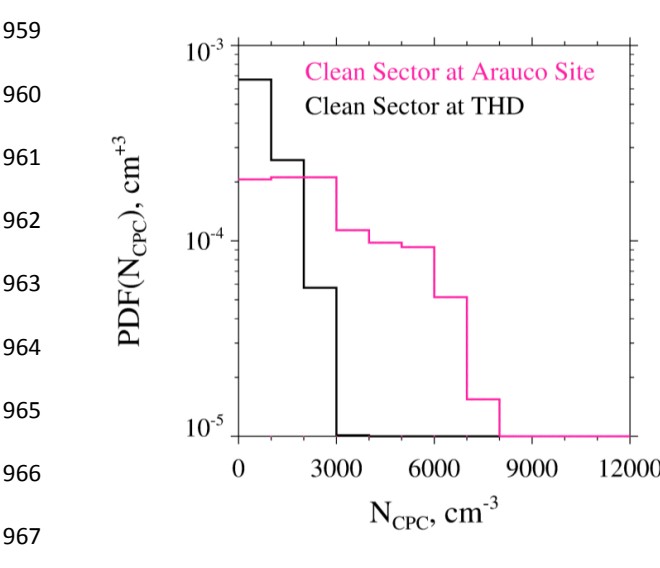

Fig. 5 - CPC concentration probability distribution functions for the Arauco Site and the
THD.



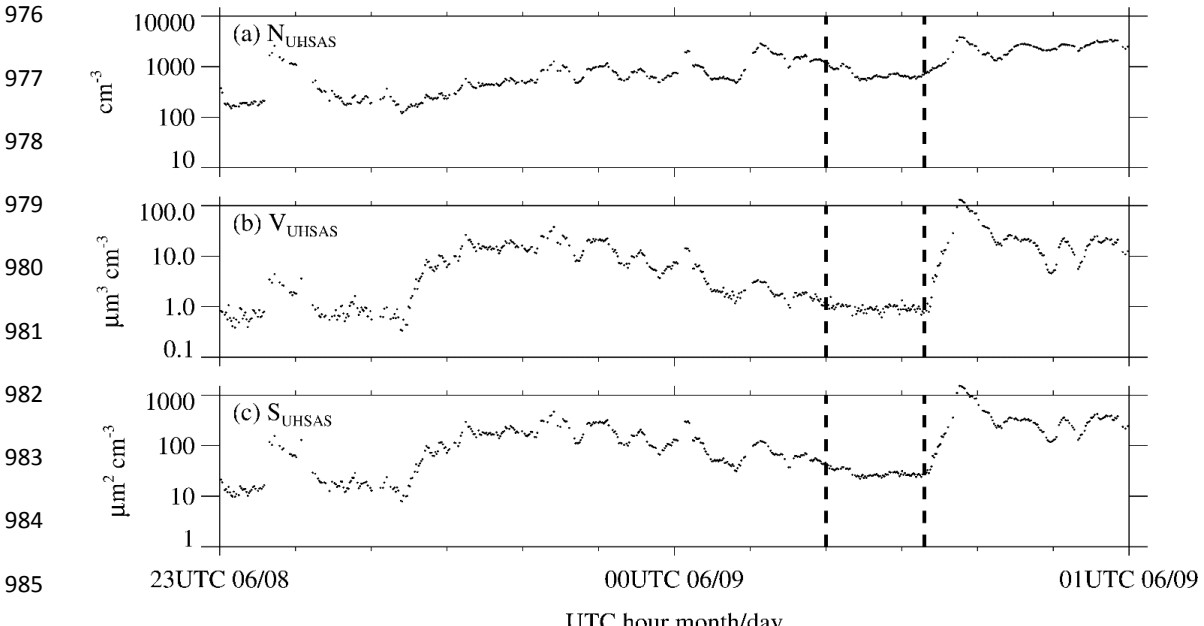

Fig. 6 – Aerosol properties centered on one of the 20 onshore trajectories that arrived at

the Arauco Site between 29 May to 28 June. This trajectory arrival occurred at 00 UTC on June

9. a) UHSAS concentration; b) UHSAS aerosol volume; c) UHSAS aerosol surface area. Aerosol

properties shown here were filtered using the procedure described in Appendix B. Vertical

dashed lines mark the subset of the two-hour segment we picked (subjectively) as being

representative of onshore-moving air that was relatively unaffected by continental aerosols

compared to adjacent portions of the two-hour segment.




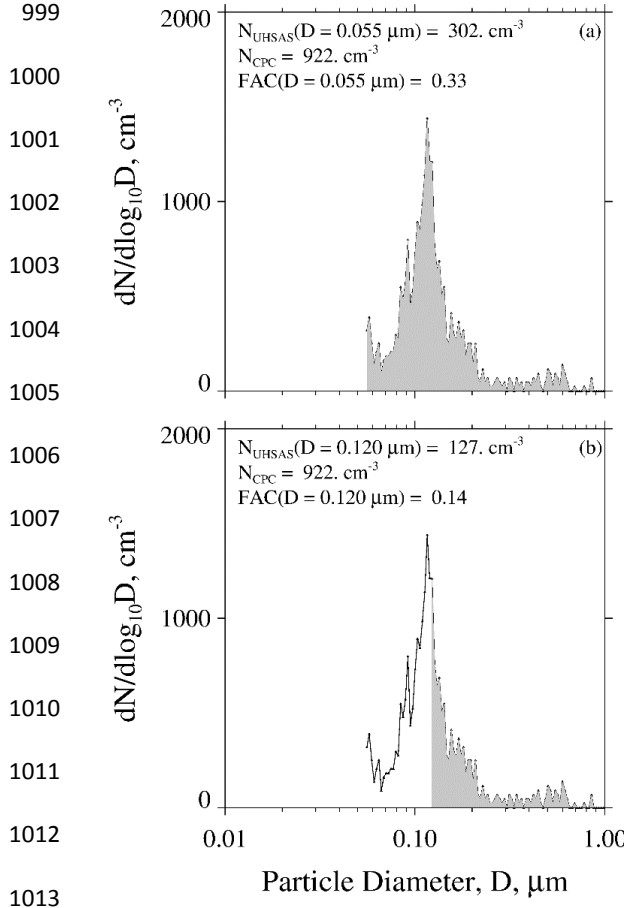



Fig. 7 - Two portrayals of the ASD recorded during CCOPE at 08:45:03 UTC June 5,
2015. This ASD is also plotted in Fig. 4a. Gray area in both panels represents the aerosol
concentration integrated from the indicated lower-limit $D$ to 1 µm. a) Figure legend has the size-
integrated UHSAS concentration, calculated with lower-limit $D$ set at 0.055 µm, the CPC
concentration, and the fractional aerosol concentration ($FAC$). b) Figure legend has the size-
integrated UHSAS concentration, calculated with lower-limit $D$ in Eq. 2 set at 0.120 µm, the
CPC concentration, and the fractional aerosol concentration ($FAC$).





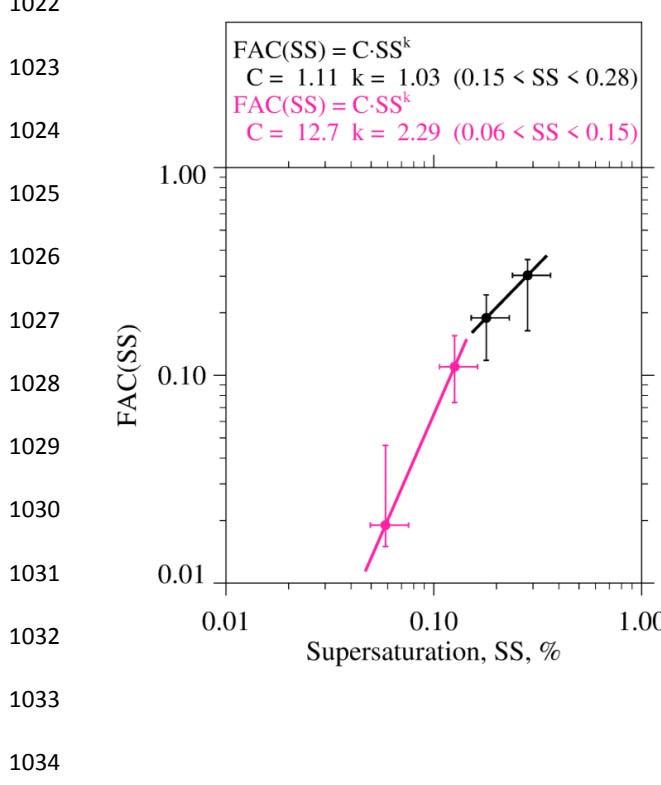

Fig. 8 - Parameterized CCN activity spectrum derived using CPC and UHSAS
measurements from the 20 onshore trajectories that arrived at the Arauco Site between 29 May
and 28 June 2015. Pink circles and the pink fit line are for lower-limit diameters set at 0.200 and
0.120 μm. Black circles and the black fit line are for lower-limit diameters set at 0.095 and 0.070
μm. Figure legend has power-law coefficients describing the parameterization; i.e., how *FAC*
varies with *SS*.



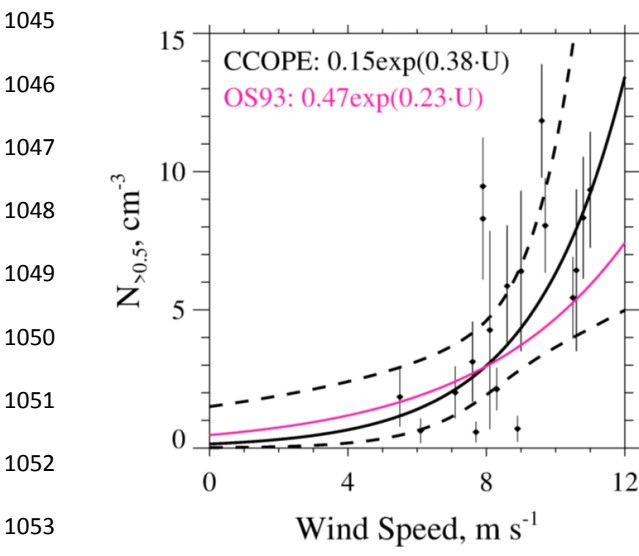

Fig. 9 – Averaged values of $N_{>0.5}$ (±1 standard deviation) vs HYSPLIT-derived averaged

$U$s for the 18 sea surface trajectories that arrived at the Arauco Site between 29 May and 28 June

2015. The black curve is the fit of the CCOPE data; dashed curves above and below the black

curves are 95% confidence intervals (Romano 1977; his Eq. 4.2.3.f). The pink curve is the fit

reported by O'Dowd and Smith (1993) for $0.38\ \mu m < D < 0.84\ \mu m$.





Fig. A1 – UHSAS internal temperature and ambient meteorological parameters at the Arauco Site over a four day period. a) Temperature inside the UHSAS; b) Temperature measured on the meteorological tower; c) *RH* measured on the meteorological tower; d) Derived *RH* inside UHSAS.



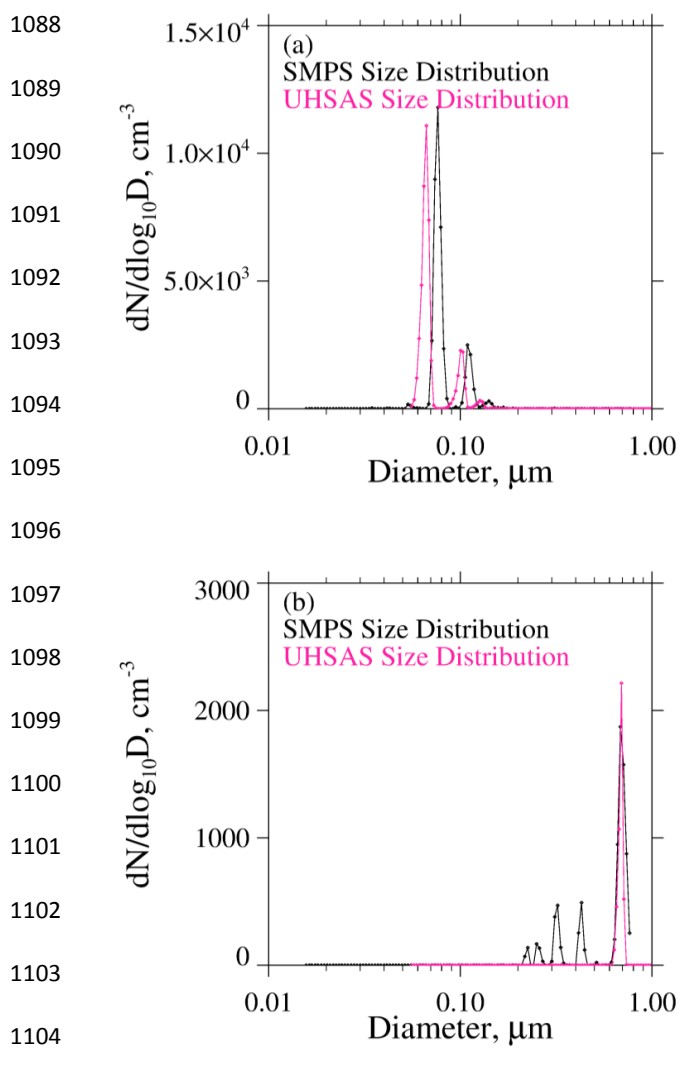

Fig. A2 – a) ASDs corresponding to mobility-selected $D = 0.075$ μm ammonium sulfate

test particles. b) ASDs corresponding to mobility-selected $D = 0.71$ μm polystyrene test

particles.





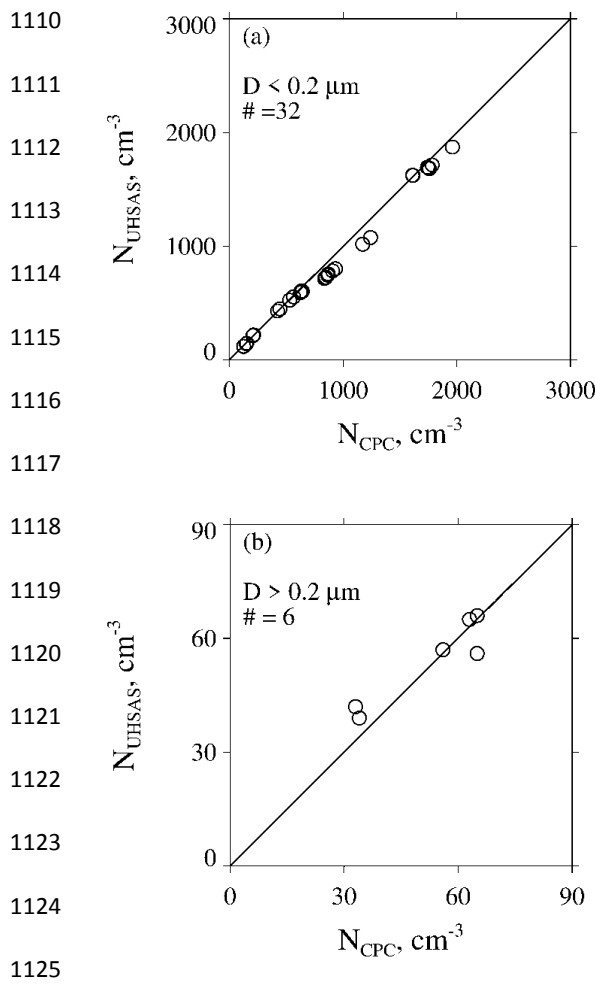

Fig. A3 - a) Size-integrated concentration from by the UHSAS versus concurrent

laboratory measurements of concentration from the CPC. Results are for mobility-selected

ammonium sulfate test particles with $D < 0.2$ μm.  b) As in Fig. A3a but for mobility-selected

ammonium sulfate test particles with $D > 0.2$ μm, and for mobility-selected polystyrene latex test

particles with $D > 0.2$ μm.





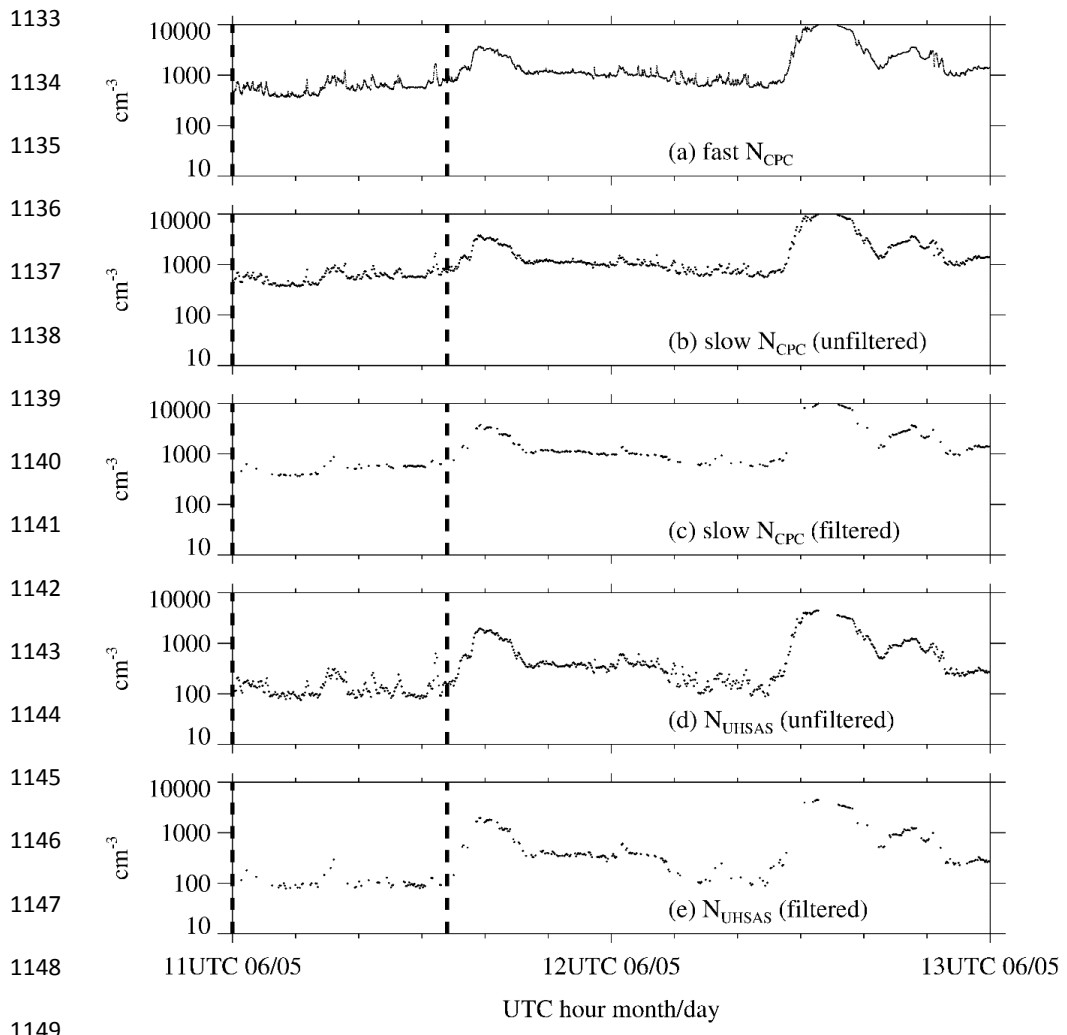

Fig. B1 - Demonstration of the numerical filter. Measurements from one of the 20

onshore trajectories that arrived at the Arauco Site between 29 May and 28 June. This trajectory

arrival occurred at 12Z June 5. a) 1-s sampled CPC measurements; b) 10-s sampled CPC

measurements; c) filtered 10-s CPC measurements; d) 10-s UHSAS measurements of size-

integrated concentration; e) filtered 10-s UHSAS measurements of size-integrated concentration.

