# Peer review of "Atmos. Chem. Phys. Discuss., https://doi.org/10.5194/acp-2019-185 Manuscript under review for journal Atmos. Chem. Phys."

_Atmospheric Chemistry and Physics, 2019_

## Referee Comment (RC1) · Anonymous Referee #2 · 13 Jun 2019

Fults et al.: Wintertime Aerosol Measurements during the Chilean Coastal Orographic Precipitation Experiment, Atmos. Chem. Phys. Discuss., https://doi.org/10.5194/acp-2019-185, in review, 2019.

**Review**

**General**
The paper presents results from a measurement campaign (CCOPE) on the Chilean Pacific Coast. The data consist of particle number concentrations measured with a condensation particle counter (CPC) and size distributions measured with a high-resolution optical particle counter (UHSAS) at a measurement station near the town of Arauco. The data are used for parameterizations of aerosol properties relevant to cloud and precipitation processes: number-to-volume ratios, concentrations of cloud condensation nuclei and sea-salt aerosol. The goal is to use these parameterizations for interpreting other data collected during the campaign on the Nahuelbuta Mountains about 30 – 100 km south of Arauco. The paper is basically well written and I can recommend its publication in ACP after some corrections and more detailed explanations.

It is in a way pleasant to see that it is still possible to make relevant observations even with such very simple traditional aerosol instrumentation when the setup in most similar campaigns today consists of several instruments measuring both physical properties and chemical composition. On the other hand, the lack of knowledge of size distributions at sizes smaller than those measured with the UHSAS, chemical composition and hygroscopicity increase the uncertainty of the interpretations. Discuss this.

The trajectories were calculated with HYSPLIT by using the GDAS wind data with a 0.5° spatial resolution. This is so coarse that the effects of local topography are not properly taken into account. The measurement site is very close to the town of Arauco and the sea, Gulf of Arauco is to the north of it and to the west of Arauco there are some hills higher than 300 m. As a result, even when the HYSPLIT trajectories show that wind blows from the west local wind in Arauco may have blown from other directions bringing anthropogenic aerosol from the town. The main goal of the paper is to use the parameterizations in the CCOPE data interpretations and modeling. During westerly winds the Nahuelbuta mountains are definitely not affected by the anthropogenic sources around the Gulf of Arauco whereas your measurement station obviously is – the average total particle number concentration in air that you classified as "clean" was $2759 \pm 1827$ cm$^{-3}$. This is high compared with marine aerosol essentially everywhere, possibly also on the coast directly to the west of the Nahuelbuta mountains. In light of this, discuss the validity of the results for CCOPE.

**Detailed comments**
Section 2.1. Add information on the distance of the Arauco measurement site from the sea, from the town of Arauco, the paper mill, the Curanilahue measurement station and the rest of the CCOPE campaign area.

L145-146 *" ... CPC concentrations were recorded once per second and once every 10 seconds (Table 1)*."
The expression "CPC concentrations" would mean there are many Condensation Particle Counters flying in the air. That is not quite correct.  Use "... CPC data were recorded..." Another thing I don't understand, is the logic of saving data once per s and once per 10 s. The 1-s data has it all, from it 10-s data can be picked up if needed. What is the logic?

The expressions "*CPC concentration*" and "*UHSAS concentration*" have been used in some sentences also later. As I wrote above, these should be rewritten. For example title of section 4.1 should rather be "Comparison of particle number concentrations..."

L256-258 " ... **194 classify as clean sector. For both sites we required a clean sector wind speed > 1.5 m s-1 in addition to the clean sector directional criteria (Fig. 2)**."
You started wind measurements at Arauco on 19 June. Did you use only the aerosol data after that in this comparison?

L286-289 " **During this two-hour data segment, centered on 00 UTC June 9 (9 pm local time), winds were light at Arauco and Curanilahue (< 2 m $s_{-1}$) and the wind direction was variable at Curanilahue (Arauco Site wind direction measurements are only available after 19 June 2015; Sect. 2.1).**"
You wrote that wind measurements at Arauco started on 19 June. How can you then write that the wind at Arauco was < 2m/s on 9 June? The distance between Arauco and Curanilahue is approximately 25 km, the measurement site of Curalinahue is at > 100 m ASL and there are quite a few valleys and hills higher than 100 m ASL between the two sites. So the local winds at these sites may have been completely different. How justifiable is it to use Curalinahue in interpreting Arauco data?

Section 4.3
In calculating the N/V ratio, justify using $N_{UHSAS}$ and not $N_{CPC}$ for N? What did HK98 and VD00 use?

Section 4.4
L377-385 This is an important part of the paper and it should be understood properly in order to understand the parameterization FAC(SS) presented later. Now it is not quite clear to me. You have earlier presented some of the simplest possible aerosol equations, Eqs. (1) – (4), which is fine, they are good to be shown. But now when it comes to a clearly more complicated issue, equations are missing which is not logical. And on line 379 it is written " **... kappa–Köhler formula of Petters and Kreidenweis (2007, their Eq. (11))**" but their Eq. (11) shows the relationship of growth factor, dry particle diameter, kappa, and relative humidity. How is this used to " **...interpret a FAC's lower-limit diameter as an upper-limit SS**" as was stated on line 377? Is the referred equation right? Write the proper equation and explain the steps of the calculation in more detail so that readers can repeat the calculation for their own data.

Section 4.5
Refer also to O'Dowd, C. D. and de Leeuw, G. (2007) and consider comparing your results also with the parameterization they presented

O'Dowd, C. D. and de Leeuw, G.: Marine Aerosol Production: a review of the current knowledge, Phil. Trans. R. Soc. A., 365,1753–1774, doi:10.1098/rsta.2007.2043, 2007

Fig 1. Add a distance scale.

Fig. 3b. Why is the y axis reverse? Why is the lowest pressure 920 hPa? A sensible scale would be 990-1020 hPa.

Fig. B1. What is the vertical dashed line at ~11:33 UTC?

---

## Referee Comment (RC2) · Anonymous Referee #1 · 28 Jun 2019

General Comments:

The paper describes aerosol data obtained in a 3-month observational study at a coastal site in Chile. Aerosol observations in this part of the world are rare so the data should be of interest to the community. Hence, I support publication of this work.

I offer some comments below that the authors can consider in revision. In general, I think some of the discussion of standard instruments and approaches could be streamlined or moved to the Appendix. The analyses and findings are fairly straightforward. Implications could be strengthened by additional comparison to observations that are clearly "clean marine".

[Figure]

Specific Comments:

Line 52: it's not clear how these aerosol indirect effects differ, as described here; please clarify. The Albrecht reference may refer to hypothesized increasing cloud lifetime and cloud cover due to increased aerosol?

Line 61: perhaps the VOCALS study should be cited as a contribution to Southern Hemisphere field work exploring aerosol-cloud interactions.

Line 70: I think you mean that the presence of SSA is associated with the presence of giant CCN that promote drizzle production.

Line 132: the particle size overestimate due to not being fully dried is discussed and a ballpark % given. However, it seems the data were not corrected for this. The CCN estimate will therefore be affected since critical supersaturation is very sensitive to size. Why wasn't this factored in? (Since a kappa is assumed, the data could be corrected for water content if RH is known.) Could this overestimate be used to add uncertainties into the parameterization?

Line 136: what height was the inlet? (this is specified only later on line 175, as 2 m) It seems to me that the aerosol inlet was much lower than is typically done for aerosol sampling campaigns (e.g., THD has an aerosol inlet at 10m). What is the impact on the data?

Line 141: there is a lot of detail about the CPC principle of operation, yet this is a very commonly applied and simple instrument. In general I think the descriptions of instrumentation could be much briefer.

Line 161: the presence of the paper mill immediately render this as a non-pristine site. Later, on lines 476, the prevalence of wood burning is mentioned. Even with onshore winds, complex coastal flows will likely result in influences from these aerosol sources. Probably it needs to be stated upfront that this site is not representative of a "clean marine" location even when data are segregated by sector.

Line 182: there is no mention of topography in the description of the site and surrounding area. This seems critical to understanding how the site is affected by transport.

Line 191: Just a comment: in the end there are only a few days (five days?) of data with onshore flow + UHSAS data that can be used to characterize the "marine" sector.

Line 231-233: I don't think these equations are needed in the text – perhaps in the supplement if you think they are necessary, but they are pretty standard.

Line 265: the T-test is a fairly standard statistical test and doesn't need a lot of description.

Line 434: internal mixing is probably not a good assumption as claimed, since many observations have shown that organics content of marine aerosol increases with decreasing size. However, it is hard to justify another assumption here, and perhaps the best way to address is to discuss some prior observations and add estimates of uncertainty?

CCN parameterization: why aren't the size distributions used more directly, and why fit with the exponential relationship? The latter is clearly not physical despite its long history of use on the community, although for marine stratus that do not reach high supersaturations, it is reasonable within the expected supersaturation bounds. What about comparing with other published spectra for coastal aerosol?

Figure 6: perhaps add local wind speed and direction to this figure?

Technical Corrections:

Line 482: "was" should be "were"

---

## Author Comment (AC1) · 14 Aug 2019

Reviewer-1

We appreciate your review and critique of the manuscript.  Thank you.

Please note: Line numbers stated here are from the original manuscript.

The paper presents results from a measurement campaign (CCOPE) on the Chilean Pacific Coast. The data consist of particle number concentrations measured with a condensation particle counter (CPC)

and size distributions measured with a high-resolution optical particle counter (UHSAS) at a measurement station near the town of Arauco. The data are used for parameterizations of aerosol properties relevant to cloud and precipitation processes: number-to-volume ratios, concentrations of cloud condensation nuclei and sea-salt aerosol. The goal is to use these parameterizations for interpreting other data collected during the campaign on the Nahuelbuta Mountains about $30 - 100$

km south of Arauco. The paper is basically well written and I can recommend its publication in ACP

after some corrections and more detailed explanations.

It is in a way pleasant to see that it is still possible to make relevant observations even with such very simple traditional aerosol instrumentation when the setup in most similar campaigns today consists of several instruments measuring both physical properties and chemical composition. On the other hand, the lack of knowledge of size distributions at sizes smaller than those measured with the UHSAS, chemical composition and hygroscopicity increase the uncertainty of the interpretations. Discuss this.

We feel that the last four paragraphs of Section 4.4 address this. Please see Section 4.4 of the revised manuscript. Finally, since estimates of the effective supersaturation (Hudson 1984) are generally smaller than 0.2%, at least in liquid-only stratocumulus (Snider et al. 2017), we do not think that lack of knowledge at $D < 0.06$ µm is a limiting factor.

Hudson, J. G., 1984: Cloud condensation nuclei measurements within clouds. J. Climate Appl.

Meteor., 23, 42–51, doi:10.1175/1520-0450(1984)023,0042:CCNMWC.2.0.CO;2.

Snider, J.R., D.Leon and Z.Wang, Droplet Concentration and Spectral Broadening in Southeast

Pacific Stratocumulus, J. Atmos. Sci., 74,  719-749, 2017

The trajectories were calculated with HYSPLIT by using the GDAS wind data with a 0.5° spatial resolution. This is so coarse that the effects of local topography are not properly taken into account. The measurement site is very close to the town of Arauco and the sea, Gulf of Arauco is to the north of it and to the west of Arauco there are some hills higher than 300 m. As a result, even when the HYSPLIT trajectories show that wind blows from the west local wind in Arauco may have blown from other directions bringing anthropogenic aerosol from the town. The main goal of the paper is to use the parameterizations in the CCOPE data interpretations and modeling. During westerly winds the Nahuelbuta Mountains are definitely not affected by the anthropogenic sources around the Gulf of Arauco whereas your measurement station obviously is – the average total particle number concentration in air that you classified as "clean" was $2759 \pm 1827$ cm$^{-3}$. This is high compared with marine aerosol essentially everywhere, possibly also on the coast directly to the west of the Nahuelbuta Mountains. In light of this, discuss the validity of the results for CCOPE.

Yes, spatial resolution of the GDAS is a factor limiting our ability to stratify measurements made at the Arauco Site. In spite of the limitation, our conditional sampling does demonstrate that aerosol surface area at the Arauco site is, on average, smaller than that reported by Hegg and Kaufman (1998) over the western Atlantic in air that had advected from the United States. The comparison of aerosol surface area is discussed in Sect. 5 of the manuscript. Related to your point about representativeness, the Arauco CPC data can be used to generate lower and upper quartile values of $N_{CPC}$ ensemble. The quartiles are 789 and 2151 cm$^{-3}$, respectively. We did not present these $N_{CPC}$ quartiles in the manuscript, but they are easily derived using the $N_{CPC}$ ensemble described in the Supplementary Material (manuscript) or using the data reader we provided (see section titled "Data Availability"). The lower quartile $N_{CPC}$ (789 cm$^{-3}$) indicates that 25% of the time conditions were comparable to the wintertime average at THD (Section 4.1).

You also assert that "..directly west of Nahuelbuta Mountains.." a more pristine aerosol state may exist. We are not convinced this is true. In Fig. 1 (revised manuscript), Lebu (population 24,000) and Cañete (population 32,000) are included. Another small city (Curanilahue) was in Figure 1 of the original manuscript. These small cities increase the possibility that cloud and precipitation over the Nahuelbuta are impacted by anthropogenic aerosols, even in a westerly flowing air. Furthermore, source/receptor relationships for aerosols on the Central Chilean Coast depend on source strength and a host of meteorological factors (e.g., extratropical cyclone track, thermal stability, and etc.).

Onshore/offshore flow that occurs during meteorologically quiescent periods (sea/land breeze circulations), could also be significant. For example, if the sea/land circulation creates a "strip" of aerosol contamination within the near-shore zone, and this air is brought onshore during episodes of persistent westerly airflow. A "coastal strip" of larger cloud droplet concentration is evident in analyses of satellite retrievals in Wood et al. (2012; their figure 4). The latter compliments the retrievals of

Bennartz (2007), who we cite in the manuscript (Sect. 5). However, neither Wood et al. (2012) nor

Bennartz (2007) segregate the satellite data into wintertime and summertime ensembles. As we state in the manuscript (Sect. 5), further analysis of the satellite retrievals are needed to investigate if the coastal strip exists both in winter and in summer.

The previous paragraph focused on aerosol-cloud interactions occurring within the planetary boundary layer; an additional dimension of the problem is aerosol resident above the planetary boundary layer. We acknowledge this in Sect. 5 (original and revised manuscript).

In summary, we feel that the caveats provided in the manuscript (Sections 5 and 6) are sufficient for numerical modelling of wintertime Chilean Coastal clouds and precipitation. We are confident that such modelling will extend understanding beyond the analyses provided here and in

Massmann et al. (2017).

Hegg, D. A., and Y. J. Kaufman, Measurements of the relationship between submicron aerosol number and volume concentration, J. Geophys. Res., 103, 5671-5678, 1998

Massmann, A.K., J.R. Minder, R.D. Garreaud, D.E. Kingsmill, R.A. Valenzuela, A. Montecinos, S.L.

Fults, and J.R. Snider, 2017, The Chilean Coastal Orographic Precipitation Experiment: Observing the

Influence of Microphysical Rain Regimes on Coastal Orographic Precipitation. J. Hydrometeor., 18,

2723–2743, https://doi.org/10.1175/JHM-D-17-0005.1, 2017

Bennartz, R., Global assessment of marine boundary layer cloud droplet number concentration from satellite, J. Geophys. Res., 112, D02201, 2007

Wood, R. ( 2006), Rate of loss of cloud droplets by coalescence in warm clouds, J.

Geophys. Res., 111, D21205, doi:10.1029/2006JD007553.

**Detailed comments**

Section 2.1. Add information on the distance of the Arauco measurement site from the sea, from the town of Arauco, the paper mill, the Curanilahue measurement station and the rest of the CCOPE

campaign area.

A distance scale is provided in Fig. 1 (revised manuscript). Also, a city Coronel (population

110,000), and two small cites Lebu (population 24,000) and Cañete (population 32,000) are included in the revised Fig. 1.

L145-146 " ... *CPC concentrations were recorded once per second and once every 10 seconds (Table 1)*."

The expression "CPC concentrations" would mean there are many Condensation Particle Counters flying in the air. That is not quite correct. Use "... CPC data were recorded..."

Corrected

Another thing I don't understand, is the logic of saving data once per s and once per 10 s. The 1-s data has it all, from it 10-s data can be picked up if needed. What is the logic?

The text was revised:

"The CPC counts particles larger than $D$ = 0.010 µm (Table 1) up to a maximum concentration of 10,000

cm$^{-3}$. The UHSAS measures scattering produced when aerosol particles are drawn through light emitted by a solid state laser (λ = 1.05 µm). By reference to a calibration table (Cai et al. 2008; Cai et al.

2013), the UHSAS microprocessor converts scattered light intensity to particle size and accumulates the derived sizes in a 99 channel histogram. Channel widths are logarithmically uniform ($\Delta log_{10}D$ =

0.013) over the instrument's full range (0.055 < $D$ < 1.0 µm). UHSAS data were recorded every 10

seconds and CPC data were recorded once per second (Table 1)."

The expressions "***CPC concentration***" and "***UHSAS concentration***" have been used in some sentences also later. As I wrote above, these should be rewritten. For example title of section 4.1 should rather be "Comparison of particle number concentrations…"

Corrected.

L256-258 " *… 194 classify as clean sector. For both sites we required a clean sector wind speed > 1.5 m*

*s-1 in* addition to the clean sector directional criteria (Fig. 2)."

You started wind measurements at Arauco on 19 June. Did you use only the aerosol data after that in this comparison?

Yes.

L286-289 " ***During this two-hour data segment, centered on 00 UTC June 9 (9 pm local time), winds were light at***

***Arauco and Curanilahue (< 2 m s-1) and the wind direction was variable at Curanilahue (Arauco Site wind direction***

***measurements are only available after 19 June 2015; Sect. 2.1).***"

You wrote that wind measurements at Arauco started on 19 June. How can you then write that the wind at Arauco was < 2m/s on 9 June?

Meteorological measurements (minus wind direction) were acquired from 29 May to 14 August and meteorological measurements (including wind direction) were acquired 19 June to 14 August. This is stated in Section 2.1 (original and revised manuscript).

The distance between Arauco and Curanilahue is approximately 25 km, the measurement site of

Curanilahue is at > 100 m ASL and there are quite a few valleys and hills higher than 100 m ASL between the two sites. So the local winds at these sites may have been completely different. How justifiable is it to use Curanilahue in interpreting Arauco data?

Reviewer #1 also commented on this, and we responded. Wind speeds were light at both locations and direction was variable at Curanilahue.  A graph of the data is provided below. In general, the effect of wind on aerosol is very difficult to interpret.

[Figure]

Section 4.3

In calculating the N/V ratio, justify using $N_{UHSAS}$ and not $N_{CPC}$ for N?

We rewrote this section of the manuscript. We feel the revision justifies what you commented on:

"In this section we analyze two ASD moments (Section 3.3). These are symbolized $N_{UHSAS}$ and $V_{UHSAS}$, respectively. The ratio of $N_{UHSAS}$ (aerosol concentration) and $V_{UHSAS}$ (aerosol volume) – generically the

*N/V* ratio - is of interest for several reasons. First, for both operational and theoretical reasons the *N/V*

ratio is evaluated for particle diameters larger than ~ 0.1 µm (VD00; Hegg and Kaufman 1998, hereafter

HK98), and importantly, the model developed to evaluate aerosol exchange between an overlying free troposphere (FT) and the marine boundary layer (MBL) successfully predicts the *N/V* ratio in the MBL

(VD00). Second, a value of the ratio can be derived by fitting measurements of *N* and *V* (HK98). Third, aerosol mass loading, and thus an aerosol volume corresponding to an assumed particle density [1], are relatively easy to evaluate. A method routinely used to evaluate aerosol mass loading involves pulling aerosol-laden air through a filter and evaluating the accumulated mass gravimetrically. Fourth, the product of an *N/V* ratio and an ambient aerosol volume (aerosol mass) has been proposed as a scheme for estimating cloud droplet concentration in marine stratocumulus clouds (HK98 and VD00).

HK98 used a passive cavity aerosol spectrometer probe (PCASP) to evaluate *N*, *V* and the *N/V* ratio.

Since the UHSAS counts down to a smaller diameter (0.055 µm) than the PCASP (0.12 µm), it is expected that the *N/V* ratios we derive using the UHSAS will be larger than those in HK98. The main reason for this is that decreasing the lower-limit diameter increases *N* more than *V* (VD00). "

Hegg, D. A., and Y. J. Kaufman, Measurements of the relationship between submicron aerosol number and volume concentration, J. Geophys. Res., 103, 5671-5678, 1998

van Dingenen, R., A. O. Virkkula, F. Raes, T. S. Bates, A. Wiedensohler, A simple non linear analytical relationship between aerosol accumulation number and sub-micron volume, explaining their observed ratio in the clean and polluted marine boundary layer, Tellus, 52B, 439-451, 2000

___________

[1] In the case of ambient particles containing hygroscopic materials, density values range between 1.5 and 1.8 g cm$^{-3}$ (McMurry et al. 2002)

What did HK98 and VD00 use?

This information is provided in Sect. 4.3. First we present $N/V$ ratios derived with the lower- limit diameter set at the minimum particle diameter detected by the UHSAS. Next, we repeat the analysis with the lower-limit diameter equal to the value applied by HK98. Results are in Tables 3 and

4.  The "headline" of these Tables provides the distinction. Additionally, VD00 integrate from minimum diameter = 0.08 μm, but we do not consider that case.

Section 4.4

L377-385 This is an important part of the paper and it should be understood properly in order to understand the parameterization FAC(SS) presented later. Now it is not quite clear to me. You have earlier presented some of the simplest possible aerosol equations, Eqs. (1) – (4), which is fine, they are good to be shown. But now when it comes to a clearly more complicated issue, equations are missing which is not logical. And on line 379 it is written " ... ***kappa–Köhler formula of Petters and Kreidenweis***

***(2007, their Eq. (11))***" but their Eq. (11) shows the relationship of growth factor, dry particle diameter, kappa, and relative humidity. How is this used to "***...interpret a FAC's lower-limit diameter as an upper-limit***

***SS***" as was stated on line 377? Is the referred equation right? Write the proper equation and explain the steps of the calculation in more detail so that readers can repeat the calculation for their own data.

The relevant equation from Petters and Kreidenweis (2007) was cited incorrectly. This is changed in the revised manuscript. For calculating critical SS, corresponding to prescribed values of dry diameter and kappa, we used Eq. 6 (Petters and Kreidenweis 2007). This is corrected in the revised manuscript. Additionally, our explanation is enhanced by inclusion of Eq. 5 (revision).

Here is the revised text:

"Our first step is to select a particle diameter, apply this as a lower-limit diameter in an integration of the UHSAS size distribution, and divide the integral by the coincident CPC-measured concentration.

The resultant is referred to as the *fractional aerosol concentration* (*FAC*).

$$FAC(D) = \frac{1}{N_{CPC}} \cdot \int_{D}^{1\,\mu m} \left( dN / d \log_{10} D \right) \cdot d \log_{10} D \qquad (5)$$

Figs. 7a - b have graphical representations of *FAC*(*D*=0.055 µm) and *FAC*(*D*=0.120 µm).

In a second step we interpret a *FAC*'s lower-limit diameter as an upper-limit *SS*. We do this by applying a value for the kappa hygroscopicity parameter, which we set at κ = 0.5, and by applying the kappa–

Köhler formula of Petters and Kreidenweis (2007, their Eq. (6)). This transformation from lower-limit *D*

to upper-limit *SS* converts the *FAC* in Fig. 7a to *FAC*(*SS* = 0.41 %) and the FAC in Fig. 7b to *FAC*(*SS* = 0.13

%). We also evaluated how a range of the kappa parameter (0.3 < κ < 0.7) translates to a range of *SS*.

Our upper-limit κ comes from airborne measurements made over the Southeast Pacific Ocean during summer (Snider et al., 2017), and our lower-limit κ is the value recommended by Andreae and

Rosenfeld (2008) for simulating aerosol indirect effects over continents."

Additionally, we rewrote the paragraph explaining how FACs are derived for onshore trajectories. The revised paragraph is this:

"The *FAC*s in Figs. 7a – b are two of the many available from CCOPE. One way to aggregate these is to calculate a *FAC* for each of the 20 onshore trajectories. For example, if we select the lower-limit diameter at $D = 0.055$ μm, plot numerator values (Eq. (5)) vs denominator values (Eq. (5)), and fit with the equation $Y = a \cdot X$, the "*a*" we derive is the *FAC(D* = 0.055 μm) for a particular trajectory. *FAC*s calculated in this way, and with lower-limit $D$ selected $= 0.120$ μm, are presented in the seventh columns of Tables 3 and 4. Correlation coefficients presented in the eighth columns of these tables mostly exceed 0.5. By averaging over the 20 onshore trajectories, we calculated the overall averages presented at the bottom of the two tables. These overall averages are *FAC(D* = 0.055 μm) = 0.35 ± 0.13

(Table 3) and *FAC(D* = 0.120 μm) = 0.13 ± 0.07 (Table 4). This decrease of the *FAC* results because a larger lower-limit $D$ (Eq. (5)), implies a smaller numerator (Eq. (5)), and thus a smaller *FAC(D)*."

Section 4.5

Refer also to O'Dowd, C. D. and de Leeuw, G. (2007) and consider comparing your results also with the parameterization they presented

O'Dowd, C. D. and de Leeuw, G.: Marine Aerosol Production: a review of the current knowledge, Phil.

Trans. R. Soc. A., 365,1753–1774, doi:10.1098/rsta.2007.2043, 2007

O'Dowd and de Leeuw (2007) summarize the sea spray research of Geever et al. (2005) and

Clarke et al. (2006). The latter two references are not compiled in Lewis and Schwartz (2004) (hereafter

LS04). We reference LS04 and Clarke et al. (2006) in the manuscript (original and revised).

Clarke et al. (2006) report a particle size-dependent flux function. As discussed in de Leeuw et al. (2011) (their section 6.5), a *size-dependent flux* can be transformed to a *concentration*, corresponding to a specified range of particle size, but this requires a steady-state, an assumed value for atmospheric residence time, and an assumed value for the depth of the MBL. Geever et al. (2005)

investigated sea spray from particles smaller than 1 $\mu$m, but did not report a size-dependent flux function.

Using the Clarke et al. (2006) parameterization with a range of wind speeds (3, 6, and 12 m/s), we transformed to concentrations assuming residence time = 3 day and MBL depth = 500 m (de Leeuw et al. (2011); their section 6.5). The SSA concentrations we calculated are within a factor = 3 of the

CCOPE curve in Fig. 9. Specifically, the calculated values are smaller at 3 m/s (Fig.9-to-calculated ratio =

1.3) and larger at 12 m/s (Fig.9-to-calculated ratio = 0.33). Given that there is significant variability in residence time and MBL depth, and in the wind speed scaling applied in Clarke et al. (2006), the result in Fig. 9 (manuscript) seems reasonable.

Summary: Because of assumptions necessary to transform a size-dependent flux to a concentration, we have not compared our result to sea spray research other than the comparison to wind-speed-dependent concentrations presented in O'Dowd and Smith (1993).

Clarke, A., V. Kapustin, S. Howell, K. Moore, B. Lienert, S. Masonis, T. Anderson, and D. Covert,

Sea-salt size distribution from breaking waves: Implications for marine aerosol production and optical extinction measurements during SEAS, J. Atmos. Ocean.Technol., 20, 1362–1374, 2003

Geever, M., C. D. O'Dowd, S. van Ekeren, R. Flanagan, E. D. Nilsson, G. de Leeuw, and Ü. Rannik,

Submicron sea spray fluxes, Geophys. Res. Lett., 32, L15810, doi:10.1029/2005GL023081, 2005

de Leeuw, G., E. L Andreas, M. D. Anguelova, C. W. Fairall, E. R. Lewis, C. O'Dowd, M. Schulz, and S. E. Schwartz, Production flux of sea spray aerosol, Rev. Geophys., 49, RG2001, doi:10.1029/2010RG000349, 2011

O'Dowd, C. and G. de Leeuw, Marine aerosol production: a review of the current knowledge,

Phil. Trans. R. Soc. A., 365,1753–1774, doi:10.1098/rsta.2007.2043, 2007

O'Dowd, C.D., and M.H. Smith, Physicochemical properties of aerosols over the Northeast

Atlantic: evidence for wind-speed-related submicron sea-salt aerosol production, J.Geophys. Res.,98,

1137-1149, 1993

Fig 1. Add a distance scale.

Fig. 1 (revised manuscript) has a distance scale.  The revised map is shown below. Small cites

Cañete and Lebu, and the city Coronel, are included in the revised Figure 1.

[Figure]

Fig. 3b. Why is the y axis reverse? Why is the lowest pressure 920 hPa? A sensible scale would be 990-

1020 hPa.

An air parcel's barometric pressure is output by the HYSPLIT model. Fig. 3b (original manuscript)

has this pressure on the Y axis.  Pressure, decreasing upward on the Y axis, is a proxy for altitude. In the revised Fig. 3b (see below), the MSL altitude of the air parcel is plotted. MSL altitude was calculated using the pressure output by HYSPLIT (parcel barometric pressure) and the ICAO equation for the

Standard Atmosphere (1993). MSL altitude increases if a larger sea-level is pressure applied in the ICAO

equation. This sensitivity is ~ 8 m / hPa.

International Civil Aviation Organization (ICAO), Manual of the ICAO Standard Atmosphere:

extended to 80 kilometres (262500 feet), 3rd ed., ISBN-92-9194-004-6, 1993

[Figure]

Fig. B1. What is the vertical dashed line at ~11:33 UTC?

This is explained in the original manuscript (Appendix B). Readers are referred to Appendix B at

L194. The first paragraph of Appendix B (revised manuscript) was revised for clarity. Here is the revised text:

"For each of the onshore trajectories (Sect. 3.1), a two-hour segment, centered on the trajectory arrival time was analyzed. An example is in Figs. B1a – e. The first panel (Fig. B1a) shows the sequence of

CPC values sampled every second (i.e., 1-s samples referred to as *fast $N_{CPC}$*), and Fig. B1b shows CPC

values sampled every 10 seconds (i.e., 10-s samples referred to as *slow $N_{CPC}$*). The following procedure was used to attenuate the narrow perturbations that were likely the result of local aerosol emissions (e.g., within the time interval indicated by vertical dashed lines in Figs. B1a, B1b, and B1d)."

---

## Author Comment (AC2) · 14 Aug 2019

Reviewer-2

We appreciate your review and critique of the manuscript. Thank you.

Please note: Line numbers stated here are from the original manuscript.

**General Comments:**

The paper describes aerosol data obtained in a 3-month observational study at a coastal site in

Chile. Aerosol observations in this part of the world are rare so the data should be of interest to the community. Hence, I support publication of this work.

I offer some comments below that the authors can consider in revision. In general, I think some of the discussion of standard instruments and approaches could be stream-lined or moved to the

Appendix.

The analyses and findings are fairly straightforward. Implications could be strengthened by additional comparison to observations that are clearly "clean marine".

This was addressed by revising the final sentences of Section 4.1:

"These averages are also statistically different (p < 0.01), and again, the Arauco average is larger than that at THD. Based on averages presented in this section, and information provided in Table

2, two summary statements are warranted: 1) During wintertime, the THD classifies as a moderately-polluted marine site, and the Arauco Site classifies between moderately-polluted marine and heavily-polluted marine. 2) These sites are not representative of conditions well removed from anthropogenic influence."

**Specific Comments:**

Line 52: it's not clear how these aerosol indirect effects differ, as described here; please clarify. The

Albrecht reference may refer to hypothesized increasing cloud lifetime and cloud cover due to increased aerosol?

We revised this:

"Consequently, upward reflection of solar radiation by liquid-only clouds (Twomey 1974), and upward reflection attributable to cloud fractional coverage (Albrecht 1989), increase with increased aerosol abundance."

Line 61: perhaps the VOCALS study should be cited as a contribution to Southern Hemisphere field work exploring aerosol-cloud interactions.

The references we picked contrast Southern and Northern Hemisphere aerosol and cloud properties.  We are not aware of a VOCALS-related publication that does that. There is reference to VOCALS in Sections 4.4 (Snider et al. 2017; manuscript bibliography).

Line 70: I think you mean that the presence of SSA is associated with the presence of giant CCN that promote drizzle production.

We do not use the modifier "giant" when referring to a subclass of the aerosol. We did change the text to stress that most of the CCN are smaller than the class of SSA particles ($D > 0.5$ um) that we focus on. Here is how the paragraph is rewritten:

"We emphasize the following topics: 1) The parameterized relationship between sea salt aerosol (SSA) particles (diameter > 0.5 µm) and wind speed; 2) The role as cloud condensation nuclei (CCN) of particles that are both smaller and more numerous than the above-mentioned SSA; 3) The parameterized relationship describing CCN activation spectra (Rogers and Yau, 1989; chapter 6), and 4) the potential application of the SSA and CCN parameterizations in numerical modelling of wintertime Southern Hemispheric clouds and precipitation. Motivating our investigation are modeling studies (Feingold et al. 1999), and analyses of field measurements (Gerber and Frick 2012), indicating that the reduction of rainfall due to increased CCN can be negated by SSA particles."

Line 132: the particle size overestimate due to not being fully dried is discussed and a ballpark %

given. However, it seems the data were not corrected for this. The CCN estimate will therefore be affected since critical supersaturation is very sensitive to size. Why wasn't this factored in? (Since a kappa is assumed, the data could be corrected for water content if RH is known.) Could this overestimate be used to add uncertainties into the parameterization?

Our analysis of the 20% particle-size overestimate is in the figure below. The pink and black data points, and their uncertainties and fit lines, are replicated from Fig. 8 (manuscript). In addition, gray circles are plotted at critical SS values corresponding to diameters 20% smaller (kappa = 0.5 is assumed). This demonstrates that a decreased lower-limit diameter, and the resultant increased fractional aerosol concentration (FAC), propagate to an insignificant departure of the perturbed data points (gray circles) from the FAC relationship in Fig. 8. Certainly, the perturbed points remain within the uncertainties described in Section 4.4. This explains why we did not factor in a 20% particle-size overestimate into our analysis of uncertainty in Fig. 8.

[Figure]

Line 136: what height was the inlet? (this is specified only later on line 175, as 2 m) It seems to me that the aerosol inlet was much lower than is typically done for aerosol sampling campaigns (e.g.,

THD has an aerosol inlet at 10m). What is the impact on the data?

Our main concern was keeping rain out of the Arauco inlet. We accomplished this by sampling below an eave on the west side of the residence at the Arauco Site (L136). In the revision, we modified the sentence starting on L174:

"An important distinction between the sampling at THD and Arauco is the above ground level (a.g.l.) height of the aerosol inlets. This is 10 and 2 m a.g.l. at THD and Arauco, respectively. We cannot state with any certainty if the lower-height sampling at Arauco made those measurements unrepresentative."

Line 141: there is a lot of detail about the CPC principle of operation, yet this is a very commonly applied and simple instrument. In general I think the descriptions of instrumentation could be much briefer.

The two paragraphs were shortened and merged. However, relevant connections to the

CPC at THD, maximum detectable concentration, and data recording were retained.

Here is the revised text:

"The CPC counts particles larger than $D = 0.010$ µm (Table 1) [1] up to a maximum concentration of

10,000 cm$^{-3}$.  The UHSAS measures scattering produced when aerosol particles are drawn through light emitted by a solid state laser ($\lambda = 1.05$ µm). By reference to a calibration table (Cai et al. 2008; Cai et al.

2013), the UHSAS microprocessor converts scattered light intensity to particle size and accumulates the derived sizes in a 99 channel histogram. Channel widths are logarithmically uniform ($\Delta log_{10}D$ =

0.013) over the instrument's full range ($0.055 < D < 1.0$ µm). UHSAS concentrations were recorded every 10 seconds and CPC concentrations were recorded once per second (Table 1)."
* * *
[1] The CPC minimum detectable diameters we report are in fact diameters that a CPC detects particles with efficiency = 50 %. The CPC detection efficiency is a steep function of particle diameter (Weidensholer et al. 1997).

Line 161: the presence of the paper mill immediately render this as a non-pristine site. Later, on lines 476, the prevalence of wood burning is mentioned. Even with onshore winds, complex coastal flows will likely result in influences from these aerosol sources. Probably it needs to be stated upfront that this site is not representative of a "clean marine" location even when data are segregated by sector.

This is stated, after relevant analysis, in two places in the original manuscript: 1) L279 to

L282, and 2) L307 to L311. We feel this is sufficient. Also, please see our reply to your General

Comment.

Line 182: there is no mention of topography in the description of the site and surrounding area. This seems critical to understanding how the site is affected by transport.

The topography is provided in Fig. 1. Also, we assert that further analysis of satellite retrievals are needed to address this outstanding issue. Please see Sect. 5 where we discuss satellite-based cloud droplet concentration retrievals in Bennartz (2007).

Line 191: Just a comment: in the end there are only a few days (five days?) of data with onshore flow + UHSAS data that can be used to characterize the "marine" sector.

As we state on L191 to L192, there are 20 onshore trajectories that overlap with the availability of UHSAS measurements. Table 3, which is discussed later in the manuscript, has the dates and times of the onshore trajectories. These occurred on seven different days in June, 2015.

Please note that the arrival times are static: 00, 06, 12, and 18 UTC.

Line 231-233: I don't think these equations are needed in the text – perhaps in the supplement if you think they are necessary, but they are pretty standard.

 Yes they are standard, however, our analysis and presentation relies on these moments (zeroth, second, and third), and our CCN parameterization relies on an integral similar to Eq. 2. We prefer to leave these definitions.

Line 265: the T-test is a fairly standard statistical test and doesn't need a lot of description.

 Apparently, there are a few tests in the category of "t-test". We prefer this one, and document by citing Havlicek and Crain (1988).

Line 434: internal mixing is probably not a good assumption as claimed, since many observations have shown that organics content of marine aerosol increases with decreasing size.   However, it is hard to justify another assumption here, and perhaps   the best way to address is to discuss some prior observations and add estimates of uncertainty?

Given that our parameterizations are aimed at multi-dimensional models of aerosol and cloud and multi-dimensional models of aerosol, cloud, and precipitation, where the mixing state in the activation scheme is nearly always "internal", we do not see merit in exploring this issue. Further, we note that aerosol dynamics calculations confirm this assumption provided coagulation (of aerosol particles) and condensation (of trace gas onto aerosol particles) has gone on for 24 hours (Fierce et al. 2017; their Figure 2). The action of coalescence scavenging (Wood et al. 2006), occurring within clouds, is ignored in the calculations of Fierce et al. (2017), and would further shorten the time needed for the internal mixing assumption to be valid. Please note, we cite Fierce et al. (2017) in this paragraph of the manuscript.

Fierce, L., N. Riemer, and T.C. Bond, Toward Reduced Representation of Mixing State for Simulating Aerosol Effects on Climate. Bull. Amer. Meteor. Soc., 98, 971–980, https://doi.org/10.1175/BAMS-D-16-0028.1, 2017

Wood, R. ( 2006), Rate of loss of cloud droplets by coalescence in warm clouds, J.

Geophys. Res., 111, D21205, doi:10.1029/2006JD007553.

CCN parameterization: why aren't the size distributions used more directly, and why fit with the exponential relationship? The latter is clearly not physical despite its long history of use on the community, although for marine stratus that do not reach high supersaturations, it is reasonable within the expected supersaturation bounds.

Size distributions are used in a manner that is direct. This is explained in the revised

Section 4.4. Our explanation is enhanced by addition of Eq. 5 (revision).

What we develop is a power-function relationship between a CCN activation spectrum and supersaturation: $N(SS) = N_{CPC} \cdot FAC(SS) = N_{CPC} \cdot C \cdot SS^k$. As is the case for all power functions relating cumulative CCN concentration ($N(SS)$) and supersaturation ($SS$), cloud droplet concentration can be calculated with the activation spectrum parameters (C and k) and with measured (or assumed) updraft velocity (e.g., Johnson 1981). Thus, an analytical link between

CCN, cloud updraft, and cloud microphysics is established. Caveats associated with this approach, and why such a calculation of droplet concentration can differ somewhat from a calculation based on a numerical parcel model, are discussed in Johnson (1981).

Johnson, D.B., 1981: Analytical Solutions for Cloud-Drop Concentration. J. Atmos. Sci.,

38, 215–218, https://doi.org/10.1175/1520-0469(1981)038<0215:ASFCDC>2.0.CO;2

What about comparing with other published spectra for coastal aerosol?

As far as we can tell, no published CCN activation spectra are available for the Central Chilean

Pacific coast (e.g., Schmale et al. 2018). Our group has published *summertime* measurements of CCN

spectra (Snider et al. 2017; their Table 2). These were acquired over the subtropical Southeast Pacific, within the summertime marine boundary layer (Snider et al. 2017; Figure 1). A comparison is shown below. Since this is an open response, we have elected to show the comparison here, but not as an addition to the manuscript. First we compare our parameterized fractional aerosol concentration (*FAC*)

function to the analysis in Andreae (2009), and then we compare CCN activation spectra.

Fig. a (see below) reproduces the parameterized *FAC* curve presented in the manuscript (Fig. 8).

As we discussed in the manuscript, this was derived using size distribution and CPC measurements (please see Eq. 5 in the revised manuscript), and using the kappa–Köhler formula of Petters and

Kreidenweis (2007, their Eq. (6)). The value $\kappa = 0.5$ is assumed for the curve we show in Fig. a. A data point derived using values in Table 2 of Andreae (2009) is also presented.  Different from our approach, the measurements Andreae (2009) analyzed are from a set of CCN(SS=0.4%) and CPC measurements.

Those measurements were acquired at a variety of locations. The locations are classified as Clean

Marine, Clean Continental, Polluted Marine, and Polluted Continental (Andreae 2009). The averaged

$N(SS=0.4\%) / N_{CPC}$ ratio for these conditions is 0.36 (Andreae 2009; their table 2). At the large *SS* end of our parameterization (Fig. a), we see reasonable agreement between with Andreae (2009).

Two activation spectra – derived as $N_{CPC} \cdot FAC(SS) = N_{CPC} \cdot C \cdot SS^k$ (Section 4.4) - are shown in Fig.

b (see below). These go with upper and lower quartile values of the $N_{CPC}$ ensemble described in the

Supplementary Material (manuscript). Also presented is the averaged CCN activation spectrum based on the 36 spectra from Table 2 of Snider et al. (2017).

At SS = 0.3 % there is consistency between the Southern Hemisphere (SH) averaged summertime spectrum (Snider et al. 2017) and SH wintertime spectrum, provided the latter is compared using the lower-quartile-$N_{CPC}$ value (see previous paragraph). However, these averaged spectra have different slopes and they therefore diverge at SS < 0.3 %. A smaller slope in the summertime setting could be due to a less prominent Aitken mode (summertime), compared to a more prominent Aiken mode (wintertime).

Although this comparison is limited, we do not see a significant discrepancy between the FAC parameterization we developed, and the approach of Andreae (2009) (Fig. a). Some discrepancy is apparent between the CCN activation spectra we derive, for relatively clean wintertime conditions, with $N_{CPC}$ = 789 $cm^{-3}$, and the averaged CCN spectrum in marine conditions over the Southeast Pacific, albeit during summer and at lower latitude. This discrepancy increases with decreasing SS. More comparison data is needed to fully validate the FAC parameterization we developed in our manuscript.

Andreae, M.O., Correlation between cloud condensation nuclei concentration and aerosol optical thickness in remote and polluted regions, Atmos. Chem. Phys, 9, 543-556, 2009

Petters, M. D., and S. M. Kreidenweis, A single parameter representation of hygroscopic growth and cloud condensation nucleus activity. Atmos. Chem. Phys., 7, 1961–1971, 2007

Schmale, J., Henning, S., Decesari, S., Henzing, B., Keskinen, H., Sellegri, K., Ovadnevaite, J., Pöhlker, M. L., Brito, J., Bougiatioti, A., Kristensson, A., Kalivitis, N., Stavroulas, I., Carbone, S., Jefferson, A., Park, M., Schlag, P., Iwamoto, Y., Aalto, P., Äijälä, M., Bukowiecki, N., Ehn, M., Frank, G., Fröhlich, R., Frumau, A., Herrmann, E., Herrmann, H., Holzinger, R., Kos, G., Kulmala, M., Mihalopoulos, N., Nenes, A., O'Dowd, C., Petäjä, T., Picard, D., Pöhlker, C., Pöschl, U., Poulain, L., Prévôt, A. S. H., Swietlicki, E., Andreae, M. O., Artaxo, P., Wiedensohler, A., Ogren, J., Matsuki, A., Yum, S. S., Stratmann, F., Baltensperger, U., and Gysel, M.: Long-term cloud condensation nuclei number concentration, particle number size distribution and chemical composition measurements at regionally representative observatories, Atmos. Chem. Phys., 18, 2853-2881, https://doi.org/10.5194/acp-18-2853-2018, 2018.

[Figure]

[Figure]

Andreae (2009) Variety of Conditions (China excluded)
Average and Quartile Range
$\langle N_{CPC} \rangle = 2215 \text{ cm}^{-3}$

........ $N_{CPC} = 2151 \text{ cm}^{-3}$ (Upper Quartile $N_{CPC}$)

——— $N_{CPC} = 789 \text{ cm}^{-3}$ (Lower Quartile $N_{CPC}$)

● Snider et al. (2017) Summer Marine
Average and Quartile Range
$\langle N_{CPC} \rangle = 463 \text{ cm}^{-3}$

Figure 6: perhaps add local wind speed and direction to this figure?

We feel the verbal description – provided in the manuscript - is adequate.  The graph is provided below, but this graph is not in the revised (or original) manuscript. In general, the effect of wind on aerosol is very difficult to interpret.

[Figure]

**Technical Corrections:**

Line 482: "was" should be "were"
Corrected